# Cellular senescence drives age-dependent hepatic steatosis

Mikolaj Ogrodnik[1], Satomi Miwa[1], Tamar Tchkonia[2], Dina Tiniakos[3,4], Caroline L. Wilson[3], Albert Lahat[5], Christopher P. Day[3,6], Alastair Burt[3,7], Allyson Palmer[2], Quentin M. Anstee[3], Sushma Nagaraja Grellscheid[5], Jan H.J. Hoeijmakers[8,9], Sander Barnhoorn[8], Derek A. Mann[3], Thomas G. Bird[10,11], Wilbert P. Vermeij[8], James L. Kirkland[2], João F. Passos[1], Thomas von Zglinicki[1] & Diana Jurk[1]

The incidence of non-alcoholic fatty liver disease (NAFLD) increases with age. Cellular senescence refers to a state of irreversible cell-cycle arrest combined with the secretion of proinflammatory cytokines and mitochondrial dysfunction. Senescent cells contribute to age-related tissue degeneration. Here we show that the accumulation of senescent cells promotes hepatic fat accumulation and steatosis. We report a close correlation between hepatic fat accumulation and markers of hepatocyte senescence. The elimination of senescent cells by suicide gene-meditated ablation of p16$^{Ink4a}$-expressing senescent cells in INK-ATTAC mice or by treatment with a combination of the senolytic drugs dasatinib and quercetin (D + Q) reduces overall hepatic steatosis. Conversely, inducing hepatocyte senescence promotes fat accumulation in vitro and in vivo. Mechanistically, we show that mitochondria in senescent cells lose the ability to metabolize fatty acids efficiently. Our study demonstrates that cellular senescence drives hepatic steatosis and elimination of senescent cells may be a novel therapeutic strategy to reduce steatosis.

[1] Newcastle University Institute for Ageing, Institute for Cell and Molecular Biosciences, Campus for Ageing and Vitality, Newcastle University, Newcastle upon Tyne NE4 5PL, UK. [2] Robert and Arlene Kogod Center on Aging, Mayo Clinic, 200 First Street SW, Rochester, Minnesota 55905, USA. [3] Institute of Cellular Medicine, Newcastle University, Newcastle upon Tyne NE2 4HH, UK. [4] Department of Pathology, Aretaieio Hospital, Medical School, National & Kapodistrian University of Athens, Athens 11528, Greece. [5] Department of Biosciences, Durham University, Durham DH1 3LE, UK. [6] Liver Unit, Newcastle upon Tyne Hospitals NHS Trust, Freeman Hospital, Newcastle upon Tyne NE7 7DN, UK. [7] The University of Adelaide, Faculty of Health Science, North Terrace, Adelaide, South Australia 5005, Australia. [8] Department of Molecular Genetics, Erasmus University Medical Center, PO Box 2040, Rotterdam 3000 CA, The Netherlands. [9] CECAD Forschungszentrum, Universität zu Köln, Joseph-Stelzmann-Straße 26, Köln 50931, Germany. [10] MRC Centre for Inflammation Research, The Queen's Medical Research Institute, University of Edinburgh, Edinburgh EH16 4TJ, UK. [11] Cancer Research UK Beatson Institute, Glasgow G61 1BD, UK. Correspondence and requests for materials should be addressed to D.J. (email: diana.jurk1@ncl.ac.uk).

Non-alcoholic fatty liver disease (NAFLD) is characterized by excess hepatic fat (steatosis) in individuals who drink little or no alcohol. NAFLD is more prevalent in older populations and it ranges from simple liver steatosis, through non-alcoholic steatohepatitis (NASH) to advanced fibrosis, cirrhosis and hepatocellular carcinoma (HCC)[1]. The mechanisms underlying this condition are not understood nor is why its prevalence increases with ageing. It has been speculated that ageing processes may promote NAFLD via different mechanisms, including adipose tissue dysfunction[2], impaired autophagy[3] and oxidative stress[4].

Cellular senescence is a state of irreversible cell-cycle arrest, which can be induced by a variety of stressors, including telomere dysfunction and genotoxic and oxidative stress[5]. Senescent cells frequently have increased secretion of a broad repertoire of proinflammatory factors, collectively known as the senescence-associated secretory phenotype, which can induce tissue dysfunction in a paracrine manner[6]. Senescent cells have mitochondrial dysfunction, with decreased oxidative phosphorylation and concomitantly increased generation of reactive oxygen species (ROS)[7,8], caused at least partly by failing mitophagy[9]. Recent studies have demonstrated that selectively eliminating senescent cells can attenuate several age-dependent disorders[10–12]. A significant fraction of hepatocytes develop a senescent phenotype during the life course of mice[13] and with age-related liver disease in humans[14]. However, the relationship between cellular senescence and liver fat accumulation remains unclear. Here we hypothesized that cellular senescence results in impaired fat metabolism and that removal of senescent cells may diminish liver steatosis.

We found a close relationship between senescence markers and fat accumulation in hepatocytes of mice fed ad libitum (AL), dietary restricted (DR) or following dietary crossover and in a small cohort of NAFLD patients. Furthermore, clearance of senescent cells by suicide gene-meditated ablation of p16[Ink4a]-expressing senescent cells in INK-ATTAC mice and a senolytic cocktail of dasatinib plus quercetin (D + Q) reduced overall hepatic steatosis in ageing, obese and diabetic mice. In contrast, hepatocyte-specific induction of senescence by a local DNA repair defect resulted in liver steatosis. Finally, we found that induction of senescence in mouse fibroblasts and hepatocytes resulted in decreased ability to metabolize fat. Our findings suggest that interventions targeting senescent cells may be developed into therapies to reduce steatosis during NAFLD.

## Results

**DR protects against liver fat deposition.** In order to investigate the relationship between fat deposition in hepatocytes and hepatocyte senescence, C57BL/6 male mice were randomly assigned to AL or DR at 3 months of age. At 12 months of age, half the animals underwent a dietary switch (crossover) for 3 months, until the age of 15 months, when all mice were killed (Fig. 1a). As shown previously[15], both long- and short-term DR were able to rescue body weight increase under AL (Supplementary Fig. 1). Interestingly, and consistent with earlier observations[15], liver weight increased in adulthood under AL conditions even faster than body weight (Fig. 1b), and this was due to hepatic fat deposition (Fig. 1c–e). Life-long DR suppressed fatty liver development (Fig. 1c–e). Importantly, short-term DR starting at middle age, reversed the increased liver mass (Fig. 1b) and liver fat accumulation (Fig. 1c–e). Contrarily, short-term return to AL after long-term DR increased body and liver weight but hepatic fat deposition remained low for at least 3 months (Fig. 1b–e). Histopathological grading confirmed progressive steatosis in AL mice and absence or minimal steatosis in DR mice (Supplementary Fig. 1b).

**DR protects against hepatocyte senescence.** Telomere dysfunction leads to activation of a persistent DNA damage response (DDR) and is a feature of cellular senescence[16]. Telomere-associated DNA damage foci (TAF, denoting co-localization of γH2A.X with a telomere PNA probe assessed by immuno-fluorescent in situ hybridization (FISH)) increase with age in mouse hepatocytes[17]. Presence of three or more TAF in a cell (Fig. 2a) is a sensitive and robust marker of senescence[18]. The frequency of hepatocytes harbouring three or more TAF increased significantly with age in AL animals but was maintained at a constant low level in DR animals (Fig. 2a,b). Both types of crossover animals (AL to DR and DR to AL) showed a significantly lower frequency of hepatocytes containing three or more TAF than AL animals, remaining at a level similar to that found in DR animals. The same pattern was observed when analysing the average number of TAF per hepatocyte but not total frequencies of DNA damage foci (Supplementary Fig. 2a,b). We next analysed the frequencies of hepatocytes harbouring another marker of cellular senescence, senescence-associated distension of satellites (SADS) (Fig. 2c). Swanson et al.[19] first reported that satellite DNA found at human and mouse centromeres unraveled from its compact state during senescence, a characteristic they designated as SADS. We found that, similarly to TAF, frequency of SADS increased with age in hepatocytes and this was largely prevented by DR (Supplementary Fig. 2c). Importantly, frequencies of hepatocytes containing ≥4 SADS (Fig. 2c,d) faithfully mirrored the pattern of TAF-positive cells (Fig. 2b). Hepatocyte senescence is also characterized by karyomegaly[20]. We analysed hepatocyte nuclear size by morphometric analysis of 4,6-diamidino-2-phenylindole (DAPI)-stained liver sections and quantified frequencies of hepatocytes with a nuclear area $> 127 \mu m^2$. Again, this senescence marker exhibited exactly the same pattern as TAF and SADS (Fig. 2e): hepatocyte karyomegaly increased with age under AL, it was maintained at low levels under long-term DR, and was reduced during late-onset short-term DR and for at least 3 months following cessation of DR. In fact, karyomegaly and SADS resulted in very similar quantitative estimates of senescent hepatocyte frequencies, while frequencies of cells with ≥3 TAFs were consistently lower (Supplementary Fig. 2d), confirming earlier results showing that between two and three TAF are necessary and sufficient to induce senescence[18,21]. Furthermore, hepatocytes sorted into fractions with normal and karyomegalic nuclei via measuring nucleus size using ImageJ showed that significantly more karyomegalic hepatocytes expressed markers of senescence, including phospho-p38 (ref. 8), 4-HNE (ref. 22), TAF and SADS, than hepatocytes with normal-sized nuclei (Supplementary Fig. 2e–h).

In order to define the impact of DR and AL dietary crossovers on senescence and fat accumulation, we conducted whole transcriptome RNA-sequencing. We then identified all genes whose expression followed the same pattern as the senescence markers across all experimental groups (Fig. 2f). Analysis of the 10 most significantly enriched (false discovery rate ≤5%) Gene Ontology (GO) categories for the genes that followed this pattern included 'lipid modification', as well as several GO terms associated with inflammatory processes, such as 'phagocytosis' and 'lymphocyte and leucocyte differentiation' (Fig. 2g).

The parallelism between senescence markers and genes involved in inflammatory responses was anticipated because senescence entails molecular reprogramming and production of a unique secretome characterized by the increased release of cytokines, chemokines, extracellular matrix remodelling factors and growth factors[23]. These factors play a role in the recruitment of immune cells, such as T cells and macrophages, which may facilitate clearance of senescent cells[24,25]. Moreover, excessive fat deposition has been associated with enhanced inflammation[26].

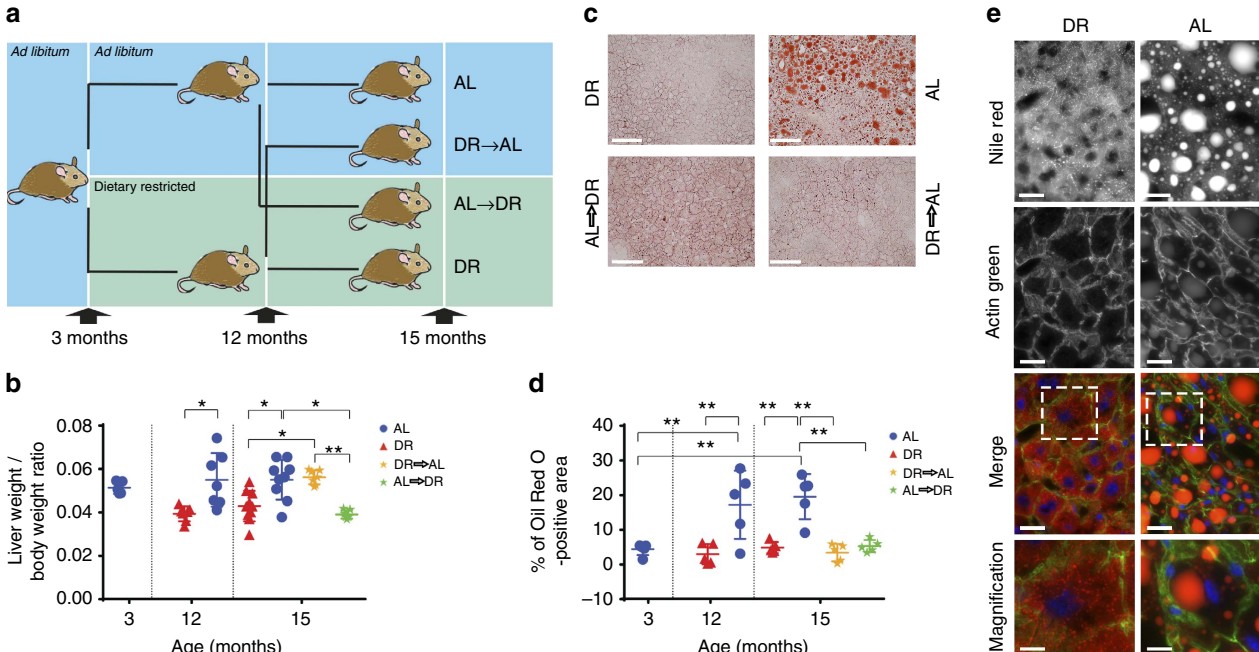

**Figure 1 | DR is protective against liver fat deposition.** (**a**) Three-month-old male mice were split into two groups and assigned to *ad libitum* (AL) or dietary restricted (DR) food supply (animals were matched by body mass and food intake). DR animals were offered 60% of AL intake as one food ration per day. After 9 months of diet (at the age of 12 months), mice were split into 4 groups liver-weight to body-weight: AL (remaining on AL feeding, $n = 9$), DR (remaining on DR feeding, $n = 10$), AL to DR (switching from AL to restricted food supply, $n = 7$), and DR to AL (switching from restricted to AL food supply, $n = 7$). Animals stayed on the assigned food regime for 3 months until killed at 15 months of age. (**b**) Liver-weight to body-weight ratios in all the experimental groups. Data per animal (dots) and means ± s.d. are shown. (**c**) Micrographs showing Oil Red O staining on frozen liver sections in 15-month-old animals in the indicated groups (red = Oil Red O, blue = haematoxylin, scale bar 100 μm). (**d**) Percentage of Oil Red O staining was determined using ImageJ ($n = 5$). (**e**) Representative micrographs showing decreased Nile red staining in DR in comparison to AL animals at 15 month of age (scale bar 20 μm, in Merge: blue = DAPI, red = Nile red, green = Actin green). Grade of steatosis was independently assessed by a liver pathologist who confirmed Oil Red O and Nile red results. All data are mean ± s.d. with 5–10 animals per group. Significant differences (one-way analysis of variance) are indicated with *$P ≤ 0.05$ and **$P ≤ 0.001$.

Consistent with this, we found by immunohistochemistry that infiltration of CD3+ and CD68+ immune cells correlated with senescent markers and fat deposition in the liver of mice under the abovementioned dietary regimes (Supplementary Fig. 2i,j).

**Senescent cell clearance decreases liver fat accumulation.** To test whether senescent cells were a cause or consequence of fat accumulation, we used two different strategies to eliminate senescent cells from older animals: (i) the INK-ATTAC mouse, in which a small molecule, AP20187 (AP) induces apoptosis through dimerization of FKBP-fused Casp8, resulting in elimination of p16-expressing cells[10,27] and (ii) the senolytic combination of the drugs D+Q, which selectively ablates senescent cells *in vitro* and *in vivo*[12]. We first used INK-ATTAC mice at 24 months of age and treated them with either AP or with D+Q and then killed the animals at 27 months of age (Fig. 3a). None of the treatments significantly altered body or liver weight (Supplementary Fig. 3a,b). As expected, untreated INK-ATTAC mice at 27 months of age displayed higher frequencies of TAF-positive (Fig. 3b) and karyomegalic (Fig. 3c) hepatocytes than control AL mice at 15 months of age (compare with Fig. 2b,e). Both AP and D+Q treatment reduced the frequencies of TAF-positive hepatocytes (Fig. 3b) and the average numbers of TAF per hepatocyte (Supplementary Fig. 3d), but total DNA damage was only significantly changed with AP but not with D+Q treatment (Supplementary Fig 3c). Moreover, AP also reduced the

percentage of karyomegalic hepatocytes, while D+Q treatment resulted in a trend towards karyomegaly reduction (Fig. 3c). Importantly, we found that both AP and D+Q administration resulted in a significant reduction in hepatic fat deposition (Fig. 3d). Furthermore, we exposed 6 month-old INK-ATTAC mice to normal chow or high-fat (HF) diet until animals were killed at 15 months of age (Fig. 3e). HF diet increased the frequency of a variety of senescent markers in hepatocytes, including TAF, karyomegaly, mRNA expression of p16 (measured by p16 and eGFP RNA-ISH) and senescence-associated β-galactosidase (SA-β-Gal) activity (Fig. 3f–k and Supplementary Fig. 3e,f). Started at 11 months of age, treatment with AP significantly reduced all analysed senescent markers in mice on HF diet (Fig. 3f–k). In accordance with the observations made in aged mice, specific elimination of senescent cells significantly reduced hepatic fat deposition in mice exposed to HF diet (Fig. 3l). Interestingly, we found across all INK-ATTAC mice (irrespectively of treatment and age) that the percentage of p16- or eGFP-positive (detected by RNA-ISH) and karyomegalic hepatocytes positively correlated with the average number of TAF (Supplementary Fig. 3g–i), which further validates its utility as a marker of senescence.

To independently confirm these results, we used db/db mice that carry a mutation in the leptin receptor gene and are a well-established model of type 2 diabetes and are characterized by liver steatosis[28]. In db/db mice, treatment with senolytic cocktail D+Q was able not only to suppress the increased fraction of TAF-containing senescent cells but also significantly reduce liver fat accumulation (Supplementary Fig. 3k–m).

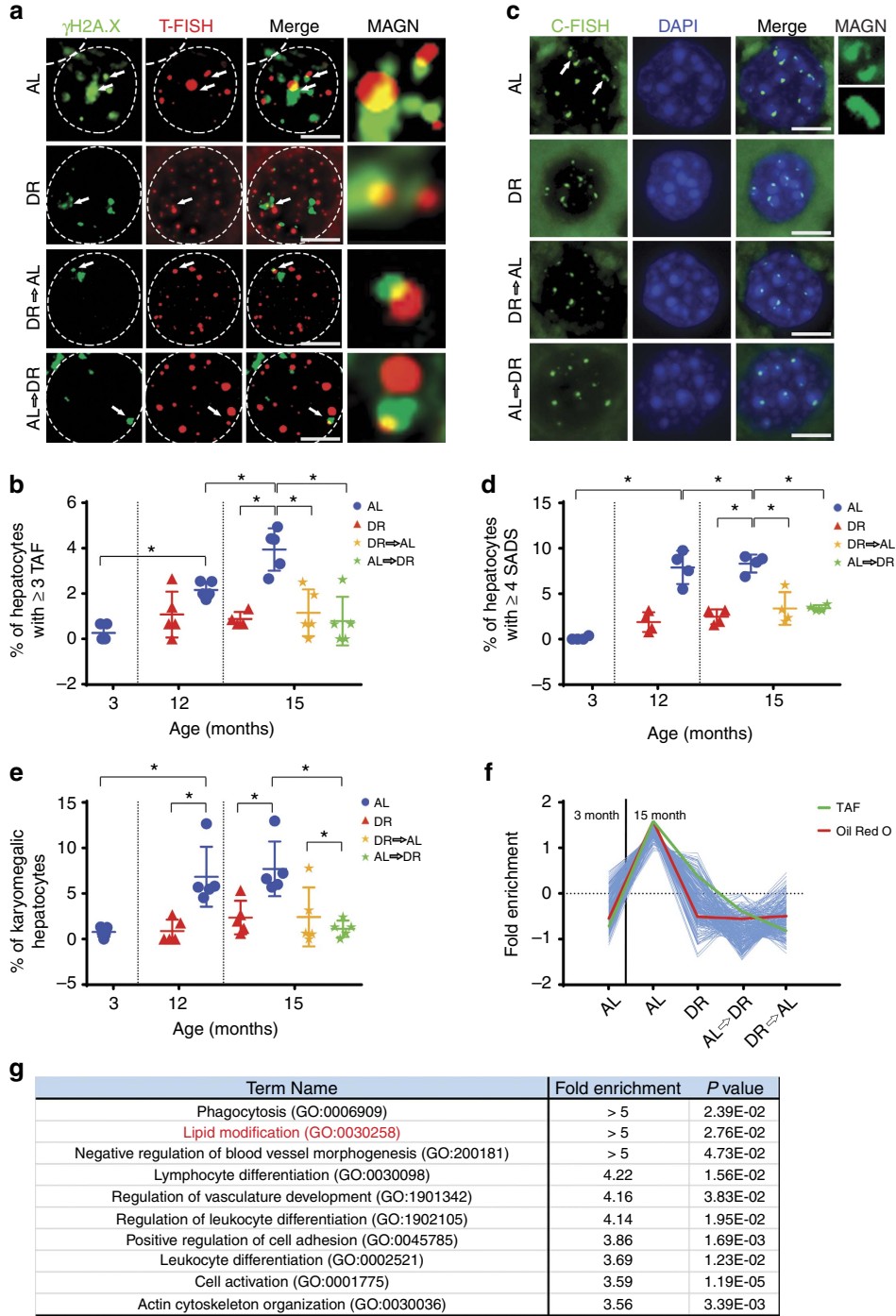

**Figure 2 | DR is protective against cellular senescence. (a)** Representative images showing γH2A.X (green) and telomere fluorescent *in situ* hybridization (FISH; red) in hepatocyte nuclei of the indicated animals at 15 months of age. White arrows indicate TAF. Panel 'MAGN' shows co-localization of γH2A.X and telomeres at higher magnification. (10 images per liver are taken containing 15–20 hepatocytes each, scale bar 4 μm). **(b)** Percentage of hepatocytes with ≥3 TAF (*n* = 5, 100–150 hepatocytes per n). **(c)** Representative images showing peri/centromeric satellite DNA signals (SADS). These signals are tightly compacted in cycling cells but distant in senescent cells. Micrographs showing single hepatocyte nuclei with centromere FISH (green) and DAPI staining (blue) in the indicated groups at 15 months of age. White arrows indicate SADS, which are shown in higher magnification at the right. (10 images per liver are taken containing 10–20 hepatocytes each, scale bar 4 μm). **(d)** Percentage of hepatocytes with ≥4 SADS (*n* = 4, 100–150 hepatocytes per *n*). **(e)** Percentage of karyomegalic hepatocytes in all experimental groups (*n* = 5, 100–140 hepatocytes per *n*). **(f)** Cluster analysis of RNA abundance as analysed by RNA-seq analysis identified 709 transcripts that follow the pattern of Oil Red O and TAF. **(g)** Transcripts identified in **f** were subjected to GO-term analysis for biological process using PANTHER. Ten pathways were over-represented (false discovery rate ≤5%), including lipid modification and inflammatory/immune system response. All data are mean ± s.d. with 4–5 animals per group. Significant differences (one-way analysis of variance) are indicated with *$P \leq 0.05$ and **$P \leq 0.001$.

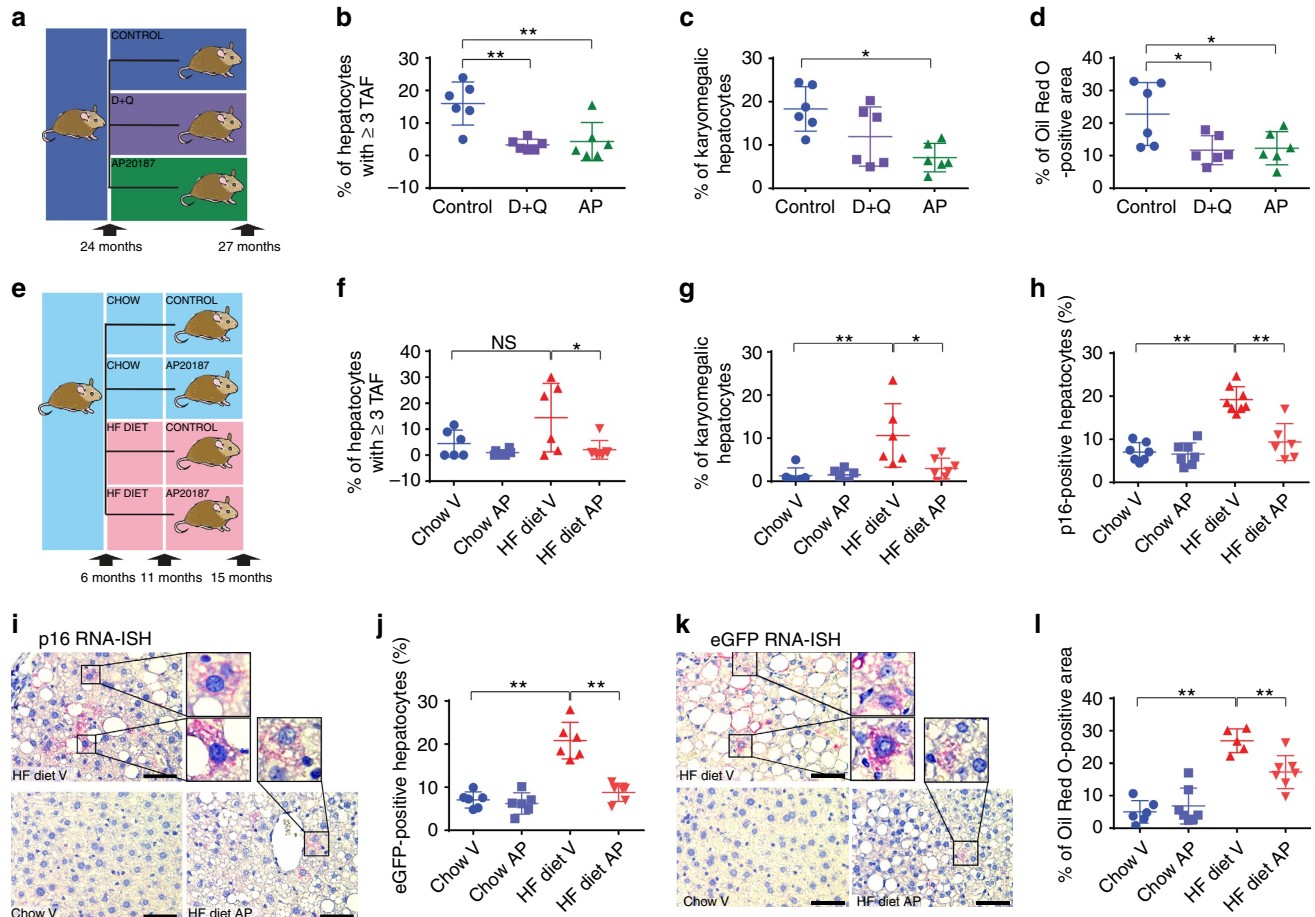

**Figure 3 | Elimination of senescent cells decreases fat accumulation in the liver.** (**a**) Twenty-four-month-old C57Bl/6[(INK-ATTAC)] male mice were split into three groups and assigned to control ($n = 6$), AP20187 ($n = 6$) or senolytic (D + Q, $n = 6$) treatment for 3 months. Percentage (**b**) of hepatocytes positive for TAF, (**c**) of karyomegalic hepatocytes and (**d**) of Oil Red O staining per area was decreased in animals treated with AP20187 and senolytic drugs at 27 months of age. (**e**) Six-month-old C57Bl/6[(INK-ATTAC)] male mice were split into four groups and assigned to chow ($n = 8$) or high fat (HF, $n = 8$) diet and treated at 11 months of age with vehicle ($n = 8$) or AP20187($n = 8$) treatment until 15 months of age. (**f**) Percentage of hepatocytes positive for ≥3 TAF ($n = 6$) and (**g**) percentage of karyomegalic hepatocytes area were significantly increased in animals fed a HF diet and significantly decreased in animals treated with AP20187 ($n = 6$-7). RNA-ISH for (**h,i**) p16 and (**j,k**) eGFP shows a significant increase in p16- and eGFP-positive hepatocytes in HF-fed mice and a significant decrease after treatment with AP20187. (**i,k**) Representative images of IHC RNA-ISH staining for eGFP and p16 in mouse liver (red/pink = p16/eGFP, blue = haematoxylin, $n = 6$-8, 10–20 images per liver). Scale bars, 100 µm. (**l**) Percentage of Oil Red O staining per area increases significantly in mice on HF diet and decreases after treatment with AP20187 ($n = 6$-7). All data are mean ± s.d. with 6–8 animals per group. Significant differences (one-way analysis of variance) are indicated with *$P \leq 0.05$ and **$P \leq 0.001$.

**Hepatocyte-specific senescence induces liver fat**. We then proceeded to test whether induction of senescence specifically in hepatocytes resulted in liver fat accumulation. For this, we generated mice with liver-specific inactivation of the DNA repair gene *Xpg*, hereafter designated *Alb-Xpg* mice, which results in accumulation of DDR markers and accelerated karyomegaly specific in hepatocytes[29]. We found that hepatocytes lacking *Xpg* over time exhibit increased markers of senescence such as TAF (Fig. 4a and Supplementary Fig. 4), karyomegaly (Fig. 4b) and p21 activation (Fig. 4c,d), which occurred concomitantly with increased age-dependent fat accumulation (Fig. 4e,f). Altogether, these data support the hypothesis that senescent cells are causally implicated in steatosis.

Cell senescence could potentially stimulate fat accumulation in a cell-autonomous fashion. Given that the efficacy of mitochondrial ATP synthesis by coupled respiration decreases during ageing[7] and in senescence[8] and that a change in mitochondrial function is indeed essential for the establishment of the senescent phenotype[30], we hypothesized that senescence might decrease

the capacity of mitochondria to oxidize fatty acids, thus contributing to fat accumulation. To test this hypothesis, we induced senescence in hepatocyte cultures isolated from young mice using X-ray irradiation, as previous studies indicated that activation of a DDR was central to the initiation of hepatocyte senescence[20,31]. Accordingly, we found that a senescent phenotype develops in cultured hepatocytes 1 week after irradiation, as shown by enhanced SA-β-Gal activity (Fig. 5a,b) and persistent DNA damage foci (Fig. 5c,d). Similarly to what we observed *in vivo*, fat droplet intensity measured by Nile red increased when comparing senescent to non-senescent hepatocytes (Fig. 5e,f). These results were confirmed independently using the lipid probe BODIPY 493/503 (Supplementary Fig. 5a–d). In order to investigate whether fat accumulation was due to impaired fatty acid oxidation, we measured cellular oxygen consumption in intact (non-permeabilized) hepatocytes. When the fatty acid palmitate was supplied as substrate, oxygen consumption increased to a lower extent in senescent hepatocytes than controls (Fig. 5g), indicating that senescent hepatocytes had indeed

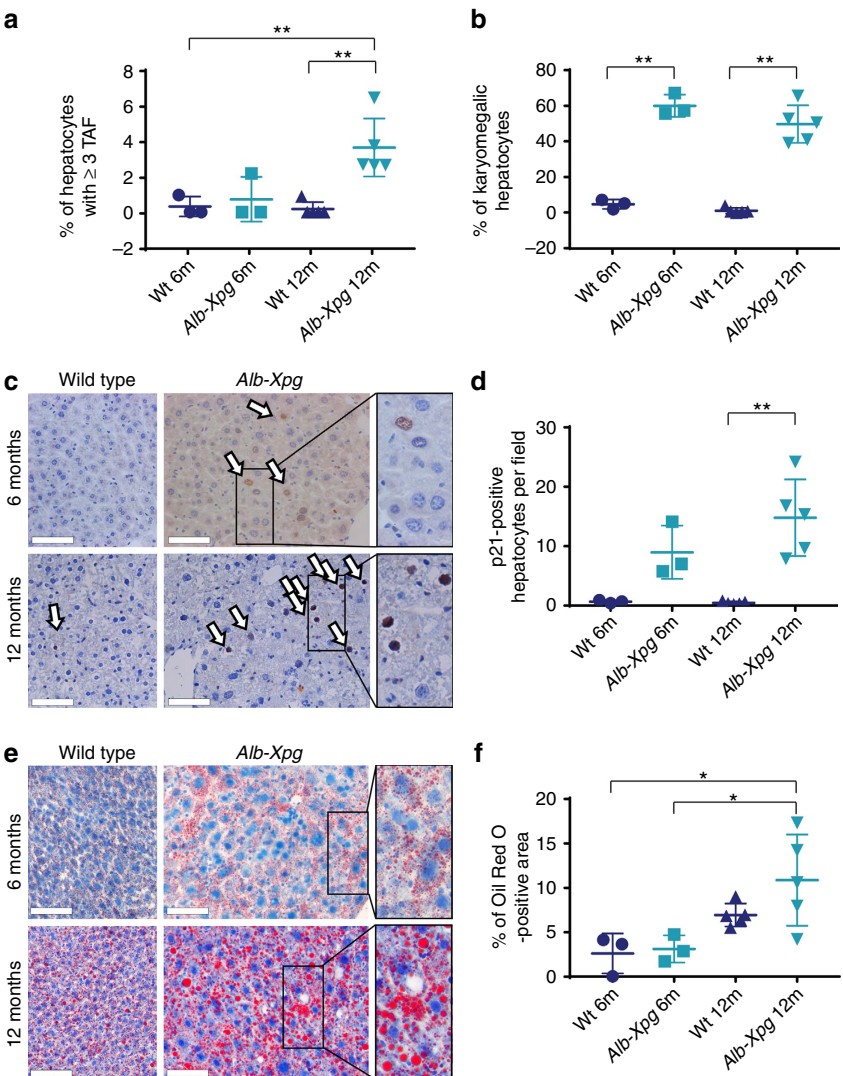

**Figure 4 | Hepatocyte-specific senescence leads to fat accumulation in the liver.** (**a**) Six ($n = 3$) and 12 month ($n = 5$) old *Alb-Xpg* mice show increased numbers of hepatocytes positive for TAF. (**b**) *Alb-Xpg* mice display increased percentage of karyomegalic hepatocytes. (**c**) Representative images of p21 IHC staining in mouse liver. (arrows indicate positive cells and area in rectangle can be seen magnified at the right, scale bar = 100 μm, brown = p21, blue = haematoxylin) (**d**) Immunohistochemistry staining shows significantly increased levels of p21-positive hepatocytes in 12-month-old *Alb-Xpg* mice. (**e**) Representative images of Oil Red O staining in the mouse liver (area in rectangle can be seen magnified at the right, scale bar = 100 μm, red = Oil Red O, blue = haematoxylin). (**f**) Percentage of Oil Red O staining per area is significantly increased in *Alb-Xpg* mice at 12 months of age. All data are mean ± s.d. with 3–5 animals per group. Significant differences (one-way analysis of variance) are indicated with *$P \leq 0.05$ and **$P \leq 0.001$.

decreased capacity to oxidize fatty acid. This was confirmed when mitochondrial fatty acid oxidation was inhibited by etomoxir. In the latter case, oxygen consumption was reduced more in controls than senescent hepatocytes (Fig. 5h). To further investigate whether this phenomenon was restricted to hepatocytes, we performed similar experiments in mouse adult fibroblasts (MAFs). Similarly to mouse hepatocytes, we found that X-ray irradiation resulted in induction of senescent markers (Supplementary Fig. 5e–i), which occurred within the same time frame as the accumulation of cytosolic fat droplets (Supplementary Fig. 5k–m). Consistent with a link between mitochondrial dysfunction and increased cytosolic fat, we found that mitochondria in senescent fibroblasts had a significantly lower respiratory control ratio (state 3/state 4 respiration rate) using the complex I-linked substrates, pyruvate and malate, resulting in reduced capacity to generate ATP by coupled respiration (Supplementary Fig. 5n,o) together with decreased capacity to oxidize fatty acids (Supplementary Fig. 5p,q). Finally,

consistent with a causal link between mitochondrial dysfunction, senescence and fat accumulation, we found that treatment of MAFs with the mitochondrial complex I inhibitor rotenone induced markers of senescence (Supplementary Fig. 5r,s) coupled with increased cytosolic fat droplet accumulation (Supplementary Fig. 5t). Together, these data indicate that senescence-associated mitochondrial dysfunction reduces cellular fatty acid oxidation capacity resulting in increased fat deposition.

**Hepatocyte senescence correlates with severity of NAFLD.** Finally, as it has been shown that telomere length decreases and DNA damage increases with steatosis grade[32], we evaluated whether TAF, which we found to be a more robust marker of senescence[17,18], and p21 in hepatocytes correlate with the severity of NAFLD. We analysed liver biopsies from nine NAFLD patients, whose demographic and histological characteristics are summarized in Table 1. Seven patients had simple steatosis or

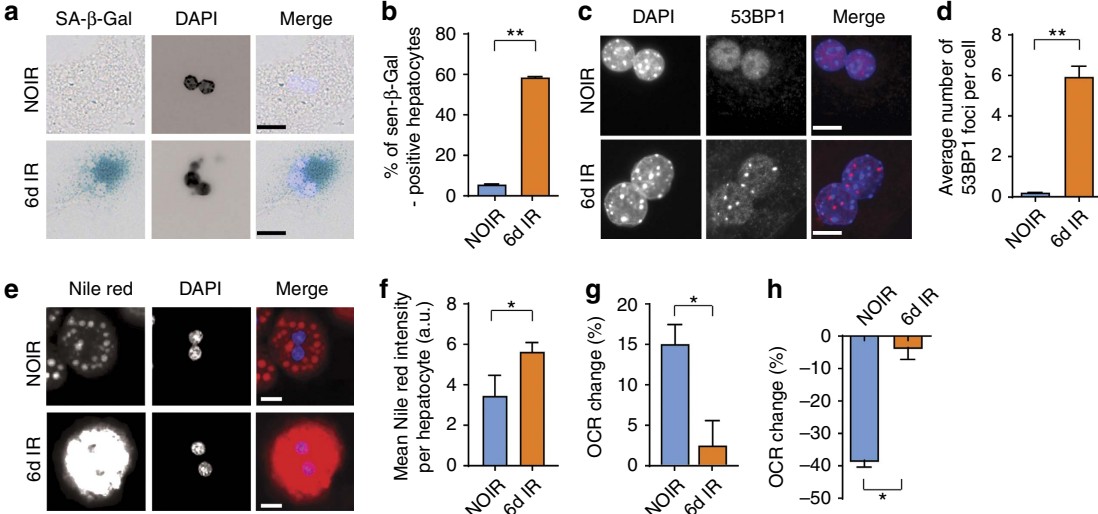

**Figure 5 | Induction of senescence in hepatocytes causes mitochondrial dysfunction and reduces fatty acid oxidation capacity.** Hepatocytes isolated from wild-type C57Bl/6 mice acquire a senescent phenotype 6 days after 10 Gy X-ray irradiation: (**a**) representative images showing SA-β-Gal in non-irradiated cells (NOIR) and 6 days after irradiation, Merge: light blue = DAPI, dark blue = SA-β-Gal, scale bar, 20 µm; (**b**) frequencies of SA-β-Gal-positive hepatocytes and (**c**) representative images showing 53BP1 staining in NOIR and 6 days after irradiation. Merge: blue = DAPI, red = 53BP1, scale bar 10 µm; (**d**) average number of 53BP1 foci per cell are significantly increased 6 days after irradiation. (**e**) Representative images showing Nile red staining in NOIR and 6 days after IR. Merge: blue = DAPI, red = Nile red, scale bar scale bar 20 µm. (**f**) Average fluorescence intensity of Nile red staining in senescent or non-senescent hepatocytes. (**g**) Change in OCR in senescent or non-senescent hepatocytes after addition of the fatty acid palmitate as substrate; (**h**) change in OCR in senescent or non-senescent hepatocytes after inhibition of fatty acid oxidation by etomoxir. Data in **b,g,h** are mean ± s.e.m. and in **d,f** are mean ± s.d. with 3–4 animals per group. Significant differences (*t*-test) are indicated with *$P \leq 0.05$ and **$P \leq 0.001$.

## Table 1 | Clinicopathological features of patients.

|  | 1 | 2 | 3 | 4 | 5 | 6 | 7 | 8 | 9 |
|---|---|---|---|---|---|---|---|---|---|
| Age at diagnosis, years | 66 | 68 | 26 | 36 | 55 | 52 | 58 | 45 | 41 |
| Gender | F | F | F | M | F | F | F | M | M |
| Steatosis grade | 1 | 2 | 1 | 1 | 2 | 2 | 3 | 1 | 3 |
| % Steatotic hepatocytes | 8 | 40 | 10 | 10 | 35 | 35 | 80 | 10 | 70 |
| Acinar zone | 3 | 2 + 3 | Non-zonal | 3 | 3 | 3 | Pan-zonal | Non-zonal | 2 + 3 |
| Hepatocyte ballooning | 0 | 1 | 0 | 0 | 0 | 0 | 1 | 0 | 0 |
| Lobular inflammation | 1 | 1 | 0 | 1 | 1 | 1 | 1 | 1 | 1 |
| Portal inflammation | 1 | 0 | 0 | 0 | 1 | 1 | 1 | 0 | 0 |
| Nuclear glycogenation | 1 | 1 | 1 | 1 | 1 | 1 | 1 | 1 | 1 |
| Megamitochondria | 0 | 1 | 0 | 0 | 1 | 1 | 0 | 0 | 0 |
| Lipogranulomas | 1 | 0 | 0 | 0 | 1 | 1 | 1 | 0 | 1 |
| SAF activity score | 1 | 2 | 0 | 1 | 1 | 1 | 2 | 1 | 1 |
| NAS score | 2 | 4 | 1 | 2 | 3 | 3 | 5 | 2 | 4 |
| Fibrosis stage | 0 | 1 | 0 | 0 | 0 | 0 | 0 | 0 | 0 |

F, female; M, male; NAS, non-alcoholic fatty liver disease (NAFLD) activity score; SAF, steatosis, activity and fibrosis.

steatosis with non-specific inflammation and two patients had steatohepatitis. We found that TAF and p21 are significantly increased in subjects with high fat content in the liver (Fig. 6a–c, Supplementary Fig. 6a). Furthermore, the percentage of TAF- and p21-positive hepatocytes (Fig. 6d,e) and the average number of TAF per cell (Supplementary Fig. 6b) positively correlates with the NAFLD activity score. Additionally, the average number of TAF- and p21-positive hepatocytes ($R = 0.4287$, $P$ value = 0.055, Supplementary Fig. 6e) and the percentage of TAF- and p21-positive hepatocytes ($R = 0.3893$, $P$ value = 0.0726, Fig. 6f) were correlated and TAF and p21 were positively related with steatosis grade (Supplementary Fig. 6c,d). Importantly, TAF correlated better than p21 with all clinical scores, confirming the potential of TAF as a robust and specific marker of senescence in human clinical settings. Interestingly, the expression of p21 was

restricted to hepatocytes and not found in any other liver cell types.

Together, our data provide evidence that senescence in hepatocytes is a major driver of liver steatosis, possibly through mitochondrial dysfunction and impaired lipid metabolism, perhaps explaining progression of NAFLD.

## Discussion

Ageing and obesity are the major risk factors for many age-related diseases, including diabetes, cancer, cardiovascular disease and NAFLD[33]. NAFLD and NASH are frequently associated with obesity, metabolic syndrome and type II diabetes[34]. It is thought that increased lipid dietary consumption, lipid synthesis or decreased lipid catabolism are major

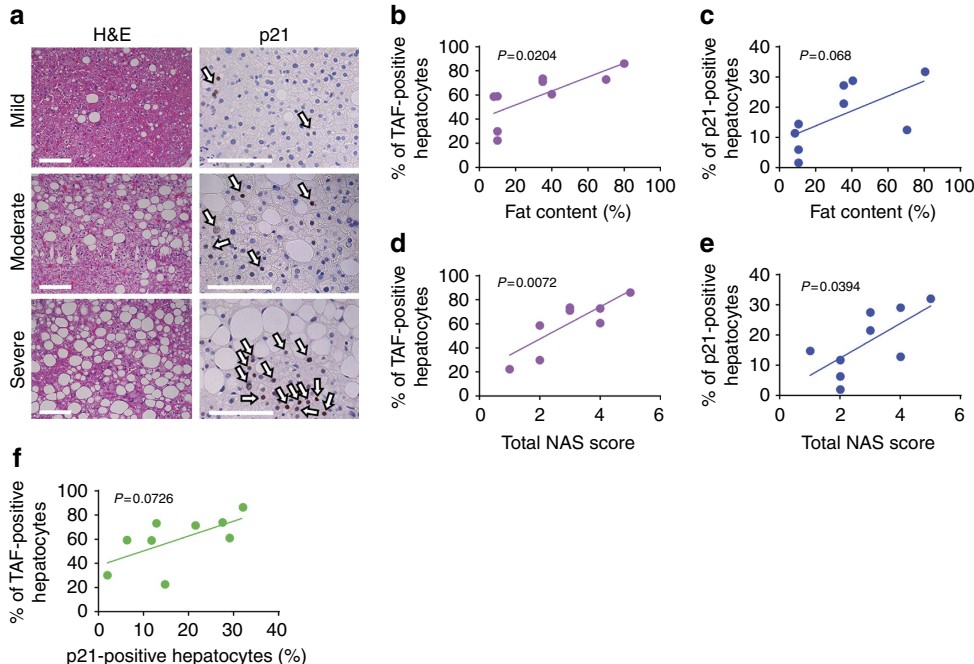

**Figure 6 | Markers of hepatocyte senescence correlate with severity of NAFLD.** (**a**) Representative images showing H&E and p21 immunohistochemistry in patients with mild, moderate and severe NAFLD (arrows indicate p21-positive cells (brown), blue = haematoxylin, scale bar, 100 μm). Mean frequencies of (**b**) TAF ($R^2 = 0.5596$) and (**c**) p21-positive hepatocytes per patient increase with the liver fat content in NAFLD patients ($R^2 = 0.3991$). Frequencies of (**d**) TAF ($R^2 = 0.6668$) and (**e**) p21-positive hepatocytes correlate with the NAS score of NAFLD patients ($R^2 = 0.4769$). (**f**) Frequencies of TAF-positive hepatocytes correlate weakly with the frequencies of p21-positive hepatocytes per patient ($R^2 = 0.3893$). All data are means (frequencies of TAF+ and p21+ cells, fat content) or scores per patient, $n = 9$.

contributors to the excessive hepatic fat observed during NAFLD[34]. Chronic steatosis has been proposed to contribute to liver injury by inducing inflammation, cell death, fibrosis and HCC[35].

Cellular senescence has been implicated in the progression of liver disease. However, the mechanisms are not yet completely understood, and in some cases, senescence has been associated with detrimental effects but in others it is not. For instance, telomere dysfunction, a major driver of hepatocyte senescence, has been shown to impair liver regeneration and induce liver cirrhosis in mice[36] but was not significantly changed during ageing in the human liver[37] and did not predict NAFLD[32]. Furthermore, chronic inflammation has been reported to drive hepatocyte senescence and contribute to liver fibrosis[18] and HCC[38]. Senescent cells have been found in the livers of NAFLD[32], cirrhotic patients[39] and in the liver of HF-fed mice[40].

In contrast, it has been suggested that p53-dependent senescence of hepatic stellate cells contributes to reduced secretion of fibrogenic proteins and is, as a result, antifibrotic[25]. However, another report suggested that activation of a proinflammatory phenotype in hepatic stellate cells could in fact drive obesity-mediated HCC[41]. Thus altogether it is still unclear if and how senescence contributes to liver dysfunction during ageing and disease.

In this study, we found that modulation of dietary intake has similar effects on hepatocyte senescence and fat accumulation during ageing in mice. Interestingly, we found that DR reduced both hepatocyte senescence and fat accumulation for a long period after its cessation in agreement with the concept of a 'metabolic memory'[15]. This led us to hypothesize a common mechanism driving both phenotypes, namely, that an impaired ability to catabolize fat could be related to the senescent phenotype.

If this was the case, strategies aimed at removing senescent cells would be viable therapeutic strategies for reducing hepatic steatosis. Consistent with this, using the INK-ATTAC mouse model in which senescent cells can be specifically deleted; we showed a reduction in liver fat content in older mice and mice exposed to HF diet, demonstrating a causal link between senescence and liver fat accumulation. We note that deletion of p16[INK4a]-expressing cells improves multiple ageing phenotypes in progeroid mice[10] and extends healthspan parameters in wild-type chronically aged mice and may increase their median lifespan[11,27,42]. In these studies, senescent cell clearance from INK-ATTAC mice was observed in multiple tissues. We therefore do not exclude the possibility that the reduced steatosis we discovered is partially a secondary outcome of clearing senescent cells from other tissues besides the liver, specifically from adipose tissue where adipocyte senescence might compromise fat storage[43]. Nonetheless, using a liver-specific mouse model of impaired DNA repair, *Alb-Xpg*[29], we found that induction of senescence specifically in hepatocytes resulted in increased fat deposition, which suggests that the effect can be cell autonomous. Moreover, using the combination of senolytic drugs, D and Q, we can effectively attain the same outcome in ageing and diabetic mice, opening the door to novel therapeutic approaches.

Our results show that cell senescence can cause steatosis cell autonomously by inducing mitochondrial dysfunction, resulting in reduced fat metabolism. Senescence-associated mitochondrial dysfunction is a regulated process driven by signalling through p21 and p38 mitogen-activated protein kinases[8] and is associated with deregulation of the mammalian target of rapamycin pathway[9,44]. This constitutes part of a central feedback loop that stabilizes the senescent phenotype[8,30]. Mitochondrial dysfunction is a feature shared by both ageing and obesity-related pathology, especially insulin resistance, and

has been associated with impaired ability to produce energy and also increases production of ROS[45]. In fact, markers of ROS-derived damage have been identified in the liver of obese individuals[46]. Furthermore, mitochondrial-derived ROS has been shown to induce telomere dysfunction, thereby contributing to cellular senescence[5]. Consistent with this, our data demonstrate that mitochondria from senescent hepatocytes cultured *in vitro* are not as effective as young hepatocytes in catabolizing the fatty acid palmitate causing accumulation of lipids in the cytosol.

In summary, our study reveals that cellular senescence drives hepatic steatosis and suggests that targeting senescent cells may be a novel pharmacological strategy to reduce steatosis.

## Methods

**Animals.** Experiments were performed in male C57Bl/6 mice aged 3, 12 and $14.2 \pm 1.2$ months[15] purchased from Harlan (Blackthorn, UK). Mice were housed in same-sex cages in groups of 4–6 ($56 \times 38 \times 18\,cm^3$, North Kent Plastics, Kent, UK) and individually identified by an ear notch. Mice were housed at $20 \pm 2\,°C$ under a 12 h light/12 h dark photoperiod with lights on at 0700 hours. The diet used was standard rodent pelleted chow (CRM (P); Special Diets Services, Witham, UK) for AL-fed mice and the same diet, but as smaller pellets, was offered to DR mice. The smaller pellet size reduced competition for food. DR mice were offered 60% of AL intake (calculated based on average food intake in 90 control AL mice between 5 and 12 months of age) as one ration at 0930 hours daily. Half of the animals were subjected to DR, while the other half, matched for body mass, food intake and age, served as AL controls. Additionally, control, young mice were killed at 3 months of age. DR was introduced at 3 months of age and lasted for 9–12 months. At the age of 12 months, some mice from the AL and DR groups had their dietary regime changed AL to DR or DR to AL for 3 months. All mice were killed at the time points mentioned above and at the end of the experiment. All work complied with the guiding principles for the care and use of laboratory animals and was licensed by the UK Home Office (PPL60/3864).

A variety of tissues were collected. Tissues were frozen in optimal cutting temperature compound or OCT media for cryosections, snap-frozen in liquid nitrogen for biochemistry and fixed in 10% formalin for 24 h before processing and paraffin embedding. Cryosectioning was performed at 10 μm intervals and paraffin-embedded tissues were cut at 3 μm intervals. Haematoxylin–eosin (H&E)-stained mouse liver sections were graded for steatosis by a single expert liver pathologist (DT) who was not aware of the genotype/treatment.

*Alb-Xpg* transgenic mice, with a liver-specific *Xpg* gene inactivation, were generated and genotyped as previously described[29]. (We used organs produced in the previous study but generated additional mice under the same conditions to increase group size.) $Xpg^{fl/−}$ $Alb-Cre^+$ mice (in a C57BL6J/FVB F1 hybrid background; referred to as Alb-Cre) are heterozygous for *Xpg* in their entire body, except for the hepatocytes in the liver, which are homozygous for *Xpg* after Cre excision of the floxed allele. Littermates, with and without Cre-recombinase expression ($Xpg^{fl/+}$ $Alb-Cre^+$ and $Xpg^{fl/−}$ $Alb-Cre^−$ respectively), were used as controls (referred to as wt). Mice were maintained in a controlled environment (20–22 °C, 12 h light; 12 h dark cycle) and were housed in individual ventilated cages under specific pathogen-free conditions. All animals had AL access to water and standard mouse food (CRM pellets, SDS BP Nutrition Ltd.; gross energy content 4.39 kcal g⁻¹ dry mass, digestible energy 3.2 kcal g⁻¹). At 6 (control: 6 male, Xpg: 6 male, 1 female) and 12 months (control: 4 male, 2 female and Xpg: 3 male, 3 female) of age, mice were killed for tissue collection. Tissues were snap-frozen in liquid nitrogen, embedded in TissueTek and sliced in 10 μm thick cryosections or fixed overnight in 10% phosphate-buffered formalin, paraffin-embedded, sectioned at 3 μm and mounted on Superfrost Plus glass slides. Oil Red O and H&E images were generated using the NanoZoomer Digital slide scanner with the NDP view software (Hamamatsu Photonics, Japan).

A new stock of INK-ATTAC transgenic mice was generated and genotyped as previously described[10]. Mice were house at 2–5 mice per cage in a 12 h light/12 h dark cycle at 24 °C with free access to food (standard mouse diet, Lab Diet 5053, St Louis, MO, USA) and water in a pathogen-free facility. AP20187 (10 mg kg⁻¹) was administered to 24-month-old mice by intraperitoneal injection every 3 days, for 3 months. For senolytic treatment, vehicle or D (5 mg kg⁻¹) and Q (10 mg kg⁻¹) in combination were administered by oral gavage once per month for 3 months. For dietary intervention studies, INK-ATTAC mice were housed 2–5 per cage, at $22 \pm 0.5\,°C$ on a 12–12-h day–night cycle and provided food and water AL. Mice were randomly assigned into the chow diet or HF diet group. HF food was purchased from Research Diets (cat no #D12492, 60% of calories in this diet are from fat). Mice were injected intraperitoneally with AP20187 (10 mg kg⁻¹) or vehicle for 3 days every 2 weeks for 10 weeks. All mice were killed at the age of 15 months (6 male HF (3 vehicle, 3 AP), 9 female HF (4 vehicle, 5 AP), 8 male control (4 vehicle, 4 AP) and 5 female control (2 vehicle, 3 AP)).

Db/db mice homozygotic males and females were purchased from Jackson Laboratory (Bar Harbor, ME, stock number: 000642). Mixed gender cohort consisting of 13 male db/db, 10 female db/db, 8 male db/+ and 8 female db/+ was first time treated at the age of 4 months. In total, four treatments

(D (5 mg kg⁻¹) and Q (50 mg kg⁻¹) or vehicle (60% Phosal, 10% ethanol and 30% PEG-400) were administered for 5 consecutive days biweekly via oral gavage.). Animals were killed at the age of 6 months.

Ethical approval was granted by the LERC Newcastle University, UK Dutch Ethical Committee at Erasmus MC (permit # 139-12-18) and the IACUC at Mayo Clinic (Protocols A26713, A40312). The work was licensed by the UK Home Office (PPL 60/3864) and complied with the guiding principles for the care and use of laboratory animals.

**Mouse adult fibroblasts.** Ear clippings were transported and stored (not longer than 1 h) in DMEM on ice. Punches were washed three times with serum-free media, finely cut and incubated for 2–3 h at 37 °C in 2 mg ml⁻¹ collagenase A in DMEM. A single-cell suspension was obtained by repeated pipetting and passing through a 24-G fine needle. Cells were centrifuged for 10 min at 1,000 r.p.m. and cultured in Advanced D-MEM/F-12 (DMEM, Invitrogen) plus 10% FCS (Sigma) in 3% $O_2$ 5% $CO_2$. Each cell strain was derived from a separate donor. MAFs were seeded and allowed to grow for 24 h and then X-ray irradiated with 5 or 10 Gy using a PXI X-Rad 225 (RPS Services Ltd) to induce cellular senescence. Alternatively, MAFs were treated with 100 nM of complex I inhibitor rotenone, which was replaced daily. Following 10 days of treatment, induction of senescent markers was observed.

**Hepatocytes.** Hepatocytes were isolated from the livers of wild-type mice by digestion with collagenase from *Clostridium histolyticum* (Sigma) and then filtered through a 70-μm cell strainer. Cells were collected by centrifugation (500 r.p.m. for 3 min), washed three times in Krebs–Ringer buffer (Sigma) and re-suspended in Williams medium E with 10% serum (WME Gibco) and plated onto collagen-coated plates (type I collagen, BD Biosciences). After 4 h, medium was removed and cells were cultured in fresh 10% or 0.5% Williams medium E. Hepatocytes were incubated at 37 °C and 3% oxygen overnight and were exposed the next day to 10 Gy irradiation in order to induce senescence. Following 10 Gy X-ray irradiation, hepatocytes acquire a morphology characteristic of senescence and SA-β-Gal activity after 6 days. Monitoring cell numbers revealed that a small percentage of hepatocytes experienced cell death after irradiation; however, most of the cells survived and acquired a senescent-like phenotype. Non-irradiated controls were analysed 1–2 days following isolation, at the same time as irradiation took place for the irradiated cells (this was necessary to prevent overgrowth of other cell types, which are present in very low numbers).

**Subjects and histological examination of liver biopsies.** Nine individuals with biopsy-proven NAFLD were evaluated. Demographic data are shown in Table 1. Liver biopsy was performed under radiological guidance. Liver tissue cores of mean length 16.3 mm, length range 9–30 mm were fixed in 10% neutral formalin and embedded in paraffin for histological examination. Tissue sections were stained with H&E and with Sirius Red Fast Green for visualizing collagen. Liver biopsies were reviewed by a single expert liver pathologist (DT) who was not aware of the clinical or immunohistochemical data. Histological diagnosis was based on currently accepted histopathological criteria for NAFLD/NASH[47]. The grade of steatosis (0–3), disease activity, including semi-quantification of lobular inflammation and hepatocyte ballooning, and stage of fibrosis (0–4) were assessed according to Kleiner et al.[48] and Bedossa et al.[49]. The NAFLD activity score (range 0–8: sum of grade of steatosis, lobular inflammation 0–3 and hepatocyte ballooning 0–2)[48] and activity according to SAF (range 0–4: sum of lobular inflammation 0–2 and hepatocyte ballooning 0–2)[49] were calculated for each biopsy.

**Oil Red O.** Preparation of Oil Red O (Sigma-Aldrich, #O1391) working solution and staining of slides was performed according to Mehlem et al.[50] and the manufacturer's instructions. Briefly, Oil Red O working solution was prepared from stock solution mixed 3:2 with water and incubated at 4 °C for 10 min. Solution was filtered through 0.45-μm filters and applied on OCT-embedded liver sections for 5 min. Slides were washed twice in water, 15 min each wash, and mounted in vectashield mounting media. For representative images, sections were counterstained with haematoxylin. Samples were imaged within 6 h. Surface of lipid droplets was quantified using the ImageJ software by measuring area occupied by red pixels.

**Nile red.** In all, 2 μl of Nile red solution (Nile red (Sigma N3013) 150 μg ml⁻¹ in acetone) were added to 1 ml 80% glycerol. Frozen OCT-embedded liver 10-μm sections were air dried for 30 min. MAFs were washed briefly with PBS and fixed for 10 min with 2% paraformaldehyde dissolved in PBS. DAPI solution was added for 10 min and afterwards sections were washed with PBS for 5 min. Some sections were stained with ActinGreen 488 (ThermoFisher, 1 drop in 0.5 ml PBS) for 30 min and washed with PBS for 3 × 5 min. In all, 20–30 μl of Nile red/glycerol were directly added to each section, mounted on a glass microscope slide and covered with a cover slip. Images were taken immediately after mounting.

**Table 2 | Nature, source and dilution of all antibodies used in the study.**

| Name of an antigen | Company producing primary antibody and catalogue number | Primary antibody: origin and concentration | Secondary antibody: origin and concentration | Tertiary antibody or developing system |
|---|---|---|---|---|
| γ-H2A.X | Cell Signalling, #9718S | Rabbit, 1:250 | Anti-rabbit, biotinylated, Goat, 1:200 | DSC-fluorescein (Vector Lab) |
| CD3 | AbD Serotec #MCA1477 | Rat, 1:100 | Anti-rat, biotinylated, Goat, 1:200 | Horseradish peroxidase ABC kit, NovaRed (Vector Lab) |
| p21 (HUGO 291) | Abcam #ab107099 | Rat, 1:250 | Anti-rat, biotinylated, Goat, 1:200 | Horseradish peroxidase ABC kit, NovaRed (Vector Lab) |
| p-p38 (Thr180/Tyr182) | Cell Signalling #4631 | Rabbit, 1:100 | Anti-rabbit, biotinylated, Goat, 1:200 | Horseradish peroxidase ABC kit, NovaRed (Vector Lab) |
| 53BP1 | Novus Biologicals #NB100-304 | Rabbit, 1:200 | Goat anti-rabbit secondary AB, Alexa Fluor 594 1:2,000 | |
| 4-HNE (HNEJ-2) | JaICA, #MHN-100P | Mouse 1:100 | Anti-mouse, biotinylated, Goat, 1:200 | Horseradish peroxidase ABC kit, NovaRed (Vector Lab) |
| CD68 | Aviva Systems Biology, #OABB00472 | Rabbit 1:250 | Anti-rabbit, biotinylated, Goat, 1:200 | Horseradish peroxidase ABC kit, NovaRed (Vector Lab) |

**BODIPY 493/503 staining.** MAFs were washed briefly with PBS and fixed for 10 min with 2% paraformaldehyde dissolved in PBS. Cells were permeabilized with PBG for 30 min and incubated for 10 min with 4 ml ml$^{-1}$ of BODIPY. Cells were washed with PBS for $3 \times 5$ min, stained with DAPI solution and mounted.

**Histochemistry and immunofluorescence (IF).** Paraffin sections were deparaffinized with Histoclear and ethanol, and antigen was retrieved by incubation in 0.01 M citrate buffer (pH 6.0) at 95 °C for 10 min. Slides were incubated in 0.9% H$_2$O$_2$ for 30 min and afterwards placed in blocking buffer (normal goat serum 1:60 in PBS/BSA, #S-1000; Vector Laboratories) for 30–60 min at room temperature (RT). Livers were further blocked with Avidin/Biotin (Vector Laboratories, no. SP-2001) for 15 min each. MAFs were washed briefly with PBS and fixed for 10 min with 2% paraformaldehyde dissolved in PBS. Cells were permeabilized for 45 min with PBG (0.5% BSA, 0.2% Fish Gelatine, 0.5% Triton X-100 in PBS). Primary antibodies were applied overnight at 4 °C. Slides were washed three times with PBS and incubated for 30 min with secondary antibody (no. PK-6101; Vector Lab). Antibodies were detected using a rabbit peroxidase ABC Kit (no. PK-6101; Vector Lab) according to the manufacturer's instructions. Substrate was developed using NovaRed (no. SK-4800; Vector Lab) or 3′3′-diaminobenzidine (no. SK4100; Vector Lab). Sections were counterstained with haematoxylin. For IF, sections were treated as before, and after the secondary antibody incubation, Fluorescein Avidin DCS (1:500 in PBS, no. A-2011, Vector Lab) was applied for 20 min. For IF on MAFs, Alexa Fluor secondary antibody (1:2,000; Molecular Probes) was applied for 30 min at RT. Sections or cells were stained with DAPI for 5–10 min and mounted in vectashield mounting media.

p21 immunohistochemistry was performed on formalin-fixed sections using rat anti-p21 antibody (clone HUGO 291H, Abcam, UK) and the ImPRESS Rat immunodetection system (MP-7444, Vector laboratories, Country) using 3′3′-diaminobenzidine (Dako, UK) as chromogen followed by counterstaining with haematoxylin. Sections were then dehydrated and coverslipped. Ten blinded consecutive non-overlapping fields were acquired at $\times 200$ magnification and quantified as previously described[51].

**Antibodies and dilutions.** The nature, source and dilution of all antibodies used in the study are listed in Table 2.

**RNA in situ hybridization.** RNA-ISH was performed after RNAscope protocol from Advanced Cell Diagnostics Inc. (ACD). Paraffin sections were deparaffinized with Histoclear, rehydrated in graded ethanol (EtOH) and H$_2$O$_2$ was applied for 10 min at RT followed by two washes in H$_2$O. Sections were placed in hot retrieval reagent and heated for 30 min. After washes in H$_2$O and 100% EtOH, sections were air dried. Sections were treated with protease plus for 30 min at 40 °C, washed with H$_2$O and incubated with target probe (p16, eGFP) for 2 h at 40 °C. Afterwards, slides were washed with H$_2$O followed by incubation with AMP1 (30 min at 40 °C) and next washed with wash buffer (WB) and AMP2 (15 min at 40 °C), WB and AMP3 (30 min at 40 °C), WB and AMP4 (15 min at 40 °C), WB and AMP5 (30 min at RT) and WB and, finally, AMP6 (15 min at RT). Finally, RNAscope 2.5 HD Reagent kit-RED was used for chromogenic labelling. After counterstaining with haematoxylin, sections were mounted and coverslipped.

**Telomere and centromere fluorescent ISH.** After γ-H2A.X IF, slides were washed three times in PBS, crosslinked with 4% paraformaldehyde for 20 min and dehydrated in graded ethanol. Sections were denatured for 10 min at 80 °C in hybridization buffer (70% formamide (Sigma UK), 25 mM MgCl$_2$, 0.1 M Tris

(pH 7.2), 5% blocking reagent (Roche, Germany)) containing 2.5 µg ml$^{-1}$ Cy-3-labelled telomere-specific (CCCTAA) or FAM-labelled, CENPB-specific (centromere) (ATTCGTTGGAAACGGGA) peptide nucleic acid probe (Panagene), followed by hybridization for 2 h at RT in the dark. Slides were washed twice with 70% formamide in $2 \times$ SSC for 15 min, followed by washes in $2 \times$ SSC and PBS for 10 min. Sections were incubated with DAPI, mounted and imaged. In-depth Z-stacking was used (a minimum of 40 optical slices with $\times 63$ objective) followed by Huygens (SVI) deconvolution. Relative telomere length was measured by telomere intensity per nucleus in one z plane.

Number of TAF per cell was assessed by quantification of partially or fully overlapping (in the same optical slice) signals from telomere probe and γ-H2A.X in slice-by-slice analysis. Number of decondensed centromeres was assessed by quantification of decondensed/elongated centromeres.

**SA-β-Gal activity.** For SA-β-Gal activity, cells were fixed with 2% paraformaldehyde for 5 min, washed and incubated at 37 °C with fresh SA-β-Gal solution: 1 mg of 5-bromo-4-chloro-3-indolyl P3-D-galactoside (X-Gal) per ml (stock $= 20$ mg of dimethylformamide per ml)/40 mM citric acid/sodium phosphate, pH 5.5/5 mM potassium ferrocyanide/5 mM potassium ferricyanide/150 mM NaCl/2 mM MgCl2. Staining was evident after 24 h. Cells were washed and stained with DAPI for 10 min, washed and mounted. For SA-β-Gal activity on the liver, 5-µm frozen sections were fixed with 0.5% glutaraldehyde for 15 min, washed with PBS and were incubated in SA-β-Gal staining solution for 18 h at 37 °C. Washed after incubation and counterstained with haematoxylin. Sections were dehydrated and mounted and 10–15 random fields were imaged per sections. Senescent hepatocytes were counted as a percentage of all hepatocytes per field.

**Karyomegaly.** In order to quantify the frequency of karyomegalic nuclei in the mouse liver precisely, nuclear staining was performed. Preparation of samples was performed as described by Wang et al.[52]. Briefly, OCT-embedded liver sections were washed three times with PBS and mounted in DAPI-containing mounting media and imaged. In-depth Z-stacking was used (a minimum of 40 optical slices with $\times 63$ objective). Analysis of karyomegaly was performed using the ImageJ software. Karyomegaly was assessed using maximum-Z projections with a threshold of 127 µm$^2$ of the nucleus area for cells to be considered karyomegalic.

**Measurements of cellular bioenergetics.** Cellular oxygen consumption rates were measured in a Seahorse XF24 Analyzer using unbuffered media (DMEM (Sigma, D-5030) supplemented with 5 mM D-glucose (Sigma), 2% L-Glutamate, 3% calf serum), and the relative changes in oxygen consumption rates after the addition of palmitate (100 µM) and etomoxir (4 µM) were calculated. Mitochondrial function in MAF was also determined in a Seahorse XF24 Analyzer (Agilent Technologies) by permeabilizing the cells using XF Plasma Membrane Permeabilizer (Agilent Technologies) according to the manufacturer's instruction using complex I-linked substrate, pyruvate (10 mM)/Malate (1 mM). The state 3 was achieved by addition of 4 mM ADP and the state 4 by oligomycin (1 µg ml$^{-1}$).

**RNA-Seq.** Strand-specific paired-end libraries for RNA-Seq were generated from DNAse-treated total RNA using Ribozero and ScriptSeq systems (Epicentre/Illumina) and run on an Illumina 2500 sequencer to obtain 100 base paired-end reads. Low quality reads were filtered out by Kraken[53]. The resulting filtered reads were mapped to the mouse genome version mm10 using Tophat[54]. Mapped reads were counted with htseq-count[55] and read counts were normalized using deseq2 (ref. 56). In order to capture genes with the same expression pattern

as the TAF and Oil Red O staining, said values were inserted into the normalized expression data set and then clustered with Biolayout express[57] using a 0.7 minimum Pearson correlation and a 95 correlation value. Clustering was conducted using MCL implementation of Markov Cluster Algorithm[58] using an inflation coefficient of 2.2 and a preinflation coefficient of 3.0. The cluster of genes with included TAF and oil lipid data were extracted and analysed for GO over-representation using the PANTHER database[59].

**Statistical analysis.** Normal distribution and equal variance were assessed using the statistical software from Sigma Plot vs11.0. We conducted one-way analysis of variance, two-tailed *t*-test and linear and nonlinear regression analysis tests using Sigma Plot v11.0 and GraphPad Prism 7.

**Ethics.** Approval was obtained from 'Newcastle and North Tyneside Research Ethics Committee' for the use of anonymized patient samples (approval reference: REC 06/Q0905/150).

**Data availability.** RNA-Seq data have been deposited in arrayexpress/ENA under accession code E-MTAB-5645. All data generated or analysed during this study are available within the paper and its Supplementary Information files and from the corresponding author on request.

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

## Acknowledgements

D.J. is funded by a Newcastle University Faculty of Medical Sciences Fellowship. C.P.D. is supported by the the National Institute for Health Research. J.F.P. is funded by a David Phillips BBSRC fellowship BB/H022384/1 and BBSRC grant BB/K017314/1. T.T., A.P. and J.L.K. were supported by NIH grant R37 AG013925 (J.L.K.), the Connor Group and the Noaber Foundation. D.A.M. is funded by Centre for Ageing & Vitality MR/M501700, MK/K001949/1 and MRC G0700890. T.G.B. is funded by a Wellcome Trust Intermediate Clinical Fellowship (WT107492). The study was supported by BBSRC (grants BB/C008200/1 and BB/I020748/1 to T.v.Z.) and grants from NUIA (D.J.) and the Newcastle Biomedical Research Centre (T.v.Z.). Financial support for J.H.J.H. was obtained from the National Institute of Health (NIH)/National Institute of Ageing (NIA) (PO1 AG017242), European Research Council Advanced Grant DamAge and Proof of Concept Grant Dementia, the KWO Dutch Cancer Society (5030), SFB628 and the Royal Academy of Arts and Sciences of the Netherlands (academia professorship to J.H.J.H.). We thank Dr Kerry Cameron for support with the mouse experiments, Professor Fiona Oakley for helpful comments and the Histology Services at the CRUK Beatson Institute for assistance with p21 immunohistochemistry.

## Author contributions

M.O. and D.J. performed the majority of experiments. S.M., T.T., D.T., C.L.W., A.B., A.L., A.P., S.N.G. and T.G.B. performed and evaluated individual experiments; Q.M.A., A.B., S.B., W.P.V., J.L.K. and C.P.D. provided materials; D.J., J.F.P. and T.v.Z. designed and supervised the study; D.J. wrote the manuscript with contributions from T.v.Z., J.L.K., D.A.M., J.H.J.H., W.P.V. and J.F.P.

## Additional information

**Competing interests:** Patents on INK-ATTAC mice and senolytic drugs are held by Mayo Clinic and licensed to Unity Biotechnology. J.L.K., A.P. and T.T. may gain financially from these patents and licenses. This research has been reviewed by the Mayo Clinic Conflict of Interest Review Board and is being conducted in compliance with Mayo Clinic conflict of interest policies. The remaining authors declare no competing financial interests.

