## [Peer Review File · Nature Communications]

Reviewer #1 (Remarks to the Author)

This report describes some very interesting data indicating that non-alcoholic fatty liver disease (NAFLD), an age-related disease phenotype, is possibly driven by cellular senescence. Senescent cells have been demonstrated to accumulate with age in various tissues, including liver, and the authors find a correlation between the accumulation of such cells and hepatic steatosis. They show that both senescent cell accumulation and steatosis are reduced in dietary restriction (DR)-treated mice. They also demonstrate that elimination of senescent cells using the INK-ATTAC mice or treatment with senolytics reduces steatosis. These are original findings and they are well presented. The methods seem sound and the manuscript is very well written. I have some general comments.

1. The authors do not really define cellular senescence in liver hepatocytes. I assume senescence in this reversed postmitotic organ occurs during regeneration. I have some difficulty in appreciating the numbers of such cells we are talking about. I can see that there could be many such cells in skin and other mitotically active tissues. In liver these numbers must be very small. Since they used the p16 marker in the INK-ATTAC mice I assume there is a way to quantify senescent cells.
2. While the thesis central to this paper, i.e., that senescence in hepatocytes is responsible for the failure of LCFA elimination due to impaired hepatic mitochondrial β -oxidation, the latter as a cause of NAFLD is not novel and has been considered previously by others.
3. The authors relate their results to obesity (in the discussion). However, they fail to mention that NAFLD is also regarded as a hepatic manifestation of metabolic syndrome and type 2 diabetes, both age-related conditions. In fact diabetes is associated with elevated fat and not only elevated glucose.
4. While the results with senescent mouse fibroblasts are interesting, they do not demonstrate a cause and effect relationship between cellular senescence and steatosis in vivo. This is still merely correlative. To do that it would be necessary to show that in senescent cells in liver in vivo fat is increased.
5. On the first Results page (there are no page numbers) the authors mention "total DNA damage". What do they mean with that and how was it measured? I did not see this in the methods.
6. I wonder why they induce senescence by radiation. Mouse fibroblasts become senescent quite spontaneously already after a few passages. Were these cells from skin of adult mice, or embryonic fibroblasts? Shouldn't they do this with liver cells? Maybe induced pluripotent stem cells obtained from liver because hepatocytes in culture do not divide.
7. The authors compare DR with ad libitum fed animals. I wonder why they did not use a high fat diet. Rats fed a HF diet develop marked steatosis and I assume this is true for mice as well. Alternatively, or in parallel, they could have used ob/ob mice, which develop spontaneous hepatic steatosis. The same is true for acyl-coenzyme A oxidase KO mice. This is the rate-limiting enzyme of peroxisomal β -oxidation of LCFA, so this could have been a way to directly test the involvement of senescent cells. In such KO mice you would not expect a relationship with senescent cells unless this condition would in fact drive cellular senescence.

Reviewer #2 (Remarks to the Author)

Below, I summarize my responses to the key points that the Editors want reviewers to address:

A. Summary of key results: The data clearly show that hepatocytes in steatotic livers are senescent and that deleting senescent cells reduces steatosis.

B. Originality and interest: The concept that hepatocytes in fatty livers are senescent is not new (over a decade ago, references demonstrated increased p21 in steatotic hepatocytes of genetically obese mice and suggested that hepatocyte senescence contributed to the impaired regenerative capacity of fatty livers). The current work does present data which support the concept that senescence could also cause hepatic steatosis (rather than simply being a consequence of steatosis). However, that evidence is indirect (i.e., acquired in cultured fibroblasts rather than hepatocytes), making it impossible to determine if the mechanisms are cell-autonomous (i.e., autocrine effects in hepatocytes) or non cell-autonomous (i.e., paracrine interactions between hepatocytes and other cell types).

C. Data and methodology: Multiple complementary approaches were used to demonstrate that hepatocytes in fatty livers are senescent and so this conclusion is assured. However, the studies that attempt to assess causality have significant limitations: a) the mechanistic experiments that assessed effects on fat metabolism were done in cultured fibroblasts rather than hepatocytes and b) the in vivo approaches used to eliminate senescent cells are not selective for hepatocytes, making it impossible to determine if hepatic steatosis improved simply because senescent hepatocytes were deleted or if the changes in liver fat reflected changes in the senescence of some other type of liver cell (e.g., liver fibroblasts) or cells outside of the liver (e.g., adipocytes).

D. Conclusions: The evidence that globally reducing senescence improves steatosis is unequivocal. However, the mechanisms involved remain obscure (see above). Also, the significance of reducing hepatic fat is uncertain in human NAFLD as natural history studies have consistently demonstrated that the severity of hepatic steatosis does not predict meaningful endpoints in human NAFLD (i.e., overall mortality, liver-related mortality, liver-related morbidity, risk of NASH, risk of liver fibrosis). These results challenge the conclusion that anti-senescent therapies would be helpful for preventing NAFLD-related morbidity/mortality in humans.

E. Suggested improvements:

- 1) Test effect of senescence on lipid metabolism in hepatocytes directly - this is necessary to conclude that senescence acts in a cell-autonomous fashion to cause hepatocyte lipid accumulation.
- 2) Determine if the secretome of fatty hepatocytes stimulates other cells/organs to produce factors that induce senescence in fatty hepatocytes. Such experiments are critical if the studies suggested in #1 yield negative results.
- 3) Quantify cell type-specific changes in various types of liver cells across the spectrum of NAFLD to determine if/how changes in senescence correlate with progression of liver injury and fibrosis. This work is necessary to judge the ultimate worth of inhibiting cellular senescence in NAFLD patients.

Reviewer #3 (Remarks to the Author)

Ogrodnik et al describe how cellular senescence acts as a driver of age-dependent hepatic steatosis. The authors analyse age-dependent hepatic steatosis and their first observation is that dietary restriction can prevent and reverse the accumulation of fat in the liver. They observe a correlation between accumulation of cells with what they describe as markers of senescence and steatosis. Using both genetic models to eliminate senescent cells and drug treatment with specificity to kill senescent cells, the authors show that ablating senescent cells results in reduced fat accumulation. By analysing RNASeq data and performing experiments with mouse fibroblasts

they suggest that mitochondria in senescent cells lose the ability to metabolise fatty acids and as a consequence accumulate lipid droplets. The study suggests that senescence is a cause driving hepatic steatosis. While this is an interesting suggestion that broadens the impact of senescence in age-associated disease, there are a number of concerns.

Major points:

1. The authors define senescent cells using as main reference the numbers of TAFs and SADs per hepatocyte, in addition they observe a correlation with 'karyomegalic hepatocytes' and p-p38 and 4HNE. While it is plausible that the authors are identifying senescent cells the markers that they use are not what any study will identify as standard markers for senescence. Ideally the authors should present data on SA-b-gal staining and p16 positive cells. While p16 staining can be technically complicated, some experiments have been performed in the INK-ATTAC mice that would allow for measurement of GFP or the INK-ATTAC transgene. There are two concerns here: that standard markers of senescence are not being used, and that the identification of SADs or TAFs in tissue sections is technically difficult, difficult to reproduce and prone to artefacts as it will require to analyse co-localisation of FISH and IF signals (or measure of distances) using confocal of the tissue slices. Also, how many nuclei have been quantified per slide? How many slides per mice?
2. The authors refer to senescence hepatocytes. Are other cell types (such as stellate cells and others) contributing to the senescence population in aged livers?
3. One of the key points of the manuscript is that the elimination of senescent cells decreases fat accumulation in liver. One of the things that is surprising on the experiment presented in Figure 3 are the numbers used and the results obtained given the known intrinsic variability of weight and fat accumulation in B1/6 mice. In the data presented in Figure 3 the size of the 3 groups analysed (control, D+Q and AP) varies between panel. For example for the control group there are 7.7.4.5 or 6 mice analysed. Why this variation? More intriguingly panel d see statistically significant differences using n=4 in the control group. Given the variation and the fact that the highest scoring mice in the control group is driving the significance I am very wary of over interpreting results that seem to be significantly underpowered.
4. Previous work from the van Deursen lab using the INK-ATTAC mice suggests that repeated treatment with AP to deplete senescent cells is successful in many organs but not in liver. The data presented here suggest that 3 single doses of AP are enough to have an effect on eliminating senescent cells in liver. These discrepancies need to be explained.
5. D+Q have been described as a senolytic drug by the Kirkland lab that participates in the study. Senolytic drugs are of great interest. While ABT-263 and ABT-737 have been used by multiple groups for eliminating senescent cells, this is not the case for D+Q. It would be interesting that the authors expand on the results to prove the efficacy of D+Q to eliminate senescent hepatocytes in the present study.
6. The authors suggest that mitochondrial dysfunction on senescent cells is responsible of reduced fatty acid oxidation capacity and explain fat accumulation on senescent cells. The authors show just correlative studies in fibroblasts and do not perform decisive manipulations to show that this and not other mechanism explain the accumulation of fat in senescent cells.
7. In the in vitro studies the authors use fibroblast. Do senescent fibroblasts (or stellate cells) accumulate in old mice showing fat deposits?
8. Senescent cells accumulate lysosome and possible other autofluorescent vesicles. Although Nile red should be a specific dye to identify lipids, additional controls or parallel methods should be used to discard artefacts.
9. The RNASeq data suggest that genes grouped under the ontology term of 'lipid modification' are upregulated during ageing. It would be interesting to link some of the specific genes identified there to the mechanism.

Point-to-point response to reviewers and editor's comments:

Comments from Editor:

- 1) Please provide a better quantification of senescent cells in the liver (reviewers #1+3), ideally distinguishing between different types of liver cells (reviewer #3)

As requested, we included a range of additional senescence markers in our analysis of liver cell senescence. We performed in addition RNA hybridisation against the senescence marker p16 (Fig.3h,i) (we used this strategy since, in our hands, antibodies against p16 do not work well in mice tissues) and against eGFP (driven by the p16 promoter in INK-ATTAC mice, Fig. 3k,l), immunohistochemistry for p21 (Fig. 4c, d) and a Senescence-associated- β -galactosidase assay (Suppl. Fig. 3f). All examined markers including the previously used TAF, SADS and karyomegaly correlated strongly with each other under the different treatments and within individual mice (Supp. Fig. 3g, h and data not shown). Importantly, all tested markers (with the sole exception of Sen- β -Gal, which is known to have low sensitivity in mouse tissues) gave very similar estimates of the total fractions of senescent hepatocytes and showed the same dependency on treatments that either eliminated senescent cells (DR, INK-ATTAC, senolytic drugs) or increased them (age, high fat feeding, db/db).

We used the specific size and morphology of hepatocytes to ensure that these data were generated exclusively from hepatocytes. We did not specifically discriminate between non-hepatocytes (endothelial cells, Kupffer cells, myofibroblasts, invading immune cells) in the livers, however, we generally observed lower expression of senescence markers in non-hepatocytes. This was true for both mouse (exemplified in the image below) and human livers, where we saw essentially complete absence of p21 staining in non-hepatocytes from NAFLD patients. The latter observation was confirmed by a pathologist (DT), who is an expert in liver disease.

Figure for review: Senescence markers in non-hepatocytes. Liver sections from 24 m old mice are shown. All telomere associated DNA damage foci (TAF, upper image) or p21-positive nuclei in the images are indicated by yellow arrows. Hepatocytes were identified by nuclear size (red dotted lines) and morphology (in the IHC image), no non-hepatocyte in the images displays senescence markers. Right panel: quantification of TAF frequencies in 24 m old livers, n=100 nuclei per group.

2) Please use hepatocytes for in vitro studies of fatty acid metabolism in senescent cells (all three reviewers)

We induced senescence in primary hepatocyte cultures isolated from young mice. Confirming the previously reported results in mouse adult fibroblasts, we now report reduced capacity to oxidise fatty acid palmitate as well as increased fat deposition in senescent hepatocytes when compared to non-senescent ones (Fig. 5).

3) Please ensure animal experiments are sufficiently powered and increase the number of animals, where needed (reviewer #3)

We made sure that sufficient animals were analysed per group and that number of animals were comparable between groups.

4) We encourage you to deplete senescent cells in an obesity model if this is feasible (eg, HFD or ob/ob; reviewer #1).

As requested, we depleted senescent cells by treatment with AP20187 in INK-ATTAC mice exposed to high-fat diet (Fig. 3e-m) and treated db/db mice (a model of diabetes and liver steatosis), which carry a mutation in the leptin receptor gene, with senolytic drugs D+Q (Supp. Fig. 3k-m). In both models, we observed that reduction of senescent hepatocytes occurred together with decreased hepatic fat accumulation. These results further highlight the therapeutic potential of targeting senescent cells to reduce steatosis.

We also think that the paper would be strengthened by further clarification of the cause-and-effect issue (see comment from reviewer #2), so please consider if there are any experiments you could do in that regard this would be. Perhaps the kill switch could be expressed (transiently) in the liver only? I would be happy to discuss this point with you.

We agree with the comments and we have highlighted in the discussion that it is plausible that the effects we observe due to elimination of senescent cells are non-cell autonomous. Unfortunately, the experiment of expressing the “kill switch” transiently in the liver would be technically challenging- particularly due to the large size of the *INK-ATTAC* construct. For that reason, we used the alternative approach of inducing senescence specifically in hepatocytes and investigating its consequences in terms of hepatic fat accumulation. We used mice with liver-specific inactivation of the DNA repair gene *Xpg* (see p. 12/13), which results in an early onset of senescence in the targeted cells (Fig. 4a-d), but does not affect any other organs or impacts on overall mortality. We observed that hepatocytes from these animals had more fat deposits (Fig. 4e, f) which is consistent with a cell-autonomous relationship between senescence and impaired fat metabolism. Moreover, experiments using adult hepatocytes isolated from mice where we induced senescence ex vivo show decreased fatty acid oxidation and increased fat deposition.

Reviewers' comments:

Reviewer #1 (Expert in ageing; Remarks to the Author):

This report describes some very interesting data indicating that non-alcoholic fatty liver disease (NAFLD), an age-related disease phenotype, is possibly driven by cellular senescence. Senescent cells have been demonstrated to accumulate with age in various tissues, including liver, and the authors find a correlation between the accumulation of such cells and hepatic steatosis. They show that both senescent cell accumulation and steatosis are reduced in dietary restriction (DR)-treated mice. They also demonstrate that elimination of senescent cells using the INK-ATTAC mice or treatment with senolytics reduces steatosis. These are original findings and they are well presented. The methods seem sound and the manuscript is very well written. I have some general comments.

We thank the reviewer for their efforts in reviewing our manuscript and for considering the work original, methodologically sound and well written and presented.

1. The authors do not really define cellular senescence in liver hepatocytes. I assume senescence in this reversed postmitotic organ occurs during regeneration. I have some difficulty in appreciating the numbers of such cells we are talking about. I can see that there could be many such cells in skin and other mitotically active tissues. In liver these numbers must be very small. Since they used the p16 marker in the INK-ATTAC mice I assume there is a way to quantify senescent cells.

We thank the reviewer for this suggestion. We measured now expression of both the native p16 and the reporter (using eGFP) in the INK-ATTAC mice. Because we (similarly to others) found p16 expression in mice difficult to assess by immunohistochemistry and immunofluorescence, we used p16 and eGFP RNA *in situ* hybridisation. This allowed us to reliably quantify positive cells (Supp. Fig. 3i and 3h-l). We found that these markers behaved very similarly under all treatments to the ones we had used before and that frequencies of p16 positive hepatocytes positively correlated with other markers of senescence when assessed in the same mice (Supp. Fig. 3g, h). Most gratifyingly, frequencies of senescent hepatocytes estimated by RNA-ISH were quantitatively very similar to those measured before by TAF and karyomegaly (Fig. 3).

Senescence of a sizeable fraction of hepatocytes in both ageing mice and humans has been shown by others and ourselves before (see papers by Aravinthan et al, Wang et al. and Kang et al cited in the ms). We do not know how this is related to the (absence of) proliferation of hepatocytes but note that multiple markers of the senescent phenotype have been observed before even in postmitotic cells and tissues.

2. While the thesis central to this paper, i.e., that senescence in hepatocytes is responsible for the failure of LCFA elimination due to impaired hepatic mitochondrial β -

oxidation, the latter as a cause of NAFLD is not novel and has been considered previously by others.

We do not claim we are the first to report the link between NAFLD, senescence and the impairment of mitochondrial β -oxidation and we did acknowledge and cite previous studies which showed and/or hypothesised these connections. However, we are the first to show that i) induction of senescence exclusively in hepatocytes drives hepatocyte fat accumulation and ii) that elimination of senescent cells, using both genetic and pharmacological approaches, rescues liver steatosis. This indicates that there is a causal relationship between hepatocyte senescence and liver steatosis in the context of liver ageing and high-fat diet.

3. The authors relate their results to obesity (in the discussion). However, they fail to mention that NAFLD is also regarded as a hepatic manifestation of metabolic syndrome and type 2 diabetes, both age-related conditions. In fact diabetes is associated with elevated fat and not only elevated glucose.

We thank the reviewer for this suggestion and now mention this connection in the discussion. Furthermore, we are currently exploring the relationship between diabetes and senescence in a separate manuscript (in preparation).

4. While the results with senescent mouse fibroblasts are interesting, they do not demonstrate a cause and effect relationship between cellular senescence and steatosis in vivo. This is still merely correlative. To do that it would be necessary to show that in senescent cells in liver in vivo fat is increased.

We agree and thank the reviewer for the insightful comment. We have generated mice with liver-specific inactivation of the DNA repair gene *Xpg*, which results in an early onset of senescence specifically in hepatocytes and observed an age-dependent increase in fat deposits (Fig. 4). Furthermore, we have induced senescence in cultured hepatocytes where we show decreased fatty acid oxidation coupled with increased fat deposits (fig. 5). Together, these results are consistent with a causal relationship between hepatocyte senescence and steatosis.

5. On the first Results page (there are no page numbers) the authors mention "total DNA damage". What do they mean with that and how was it measured? I did not see this in the methods.

We apologize for the use of this 'short-hand notion' for "total (i.e. both telomere-associated and non-telomeric) frequencies of DNA damage foci". The term has been corrected (p 5, line 1).

6. I wonder why they induce senescence by radiation. Mouse fibroblasts become senescent quite spontaneously already after a few passages. Were these cells from skin of adult mice, or embryonic fibroblasts? Shouldn't they do this with liver cells? Maybe

induced pluripotent stem cells obtained from liver because hepatocytes in culture do not divide.

We did not use the model of spontaneous arrest of mouse fibroblasts because of severe drawbacks: i) mouse fibroblasts arrest spontaneously only at un-physiologically high (21%) oxygen concentration, and this form of 'culture shock' is significantly different from human cell replicative or stress-induced senescence; ii) under this condition, the MAF genome is not very stable, leading to a high frequency of spontaneous immortalisation; and iii) mouse fibroblasts (expressing telomerase) do not spontaneously senesce at physiological oxygen partial pressure, but such IR-induced arrest mimics human senescence very well (see for instance Parinello et al. *Nature Cell Biol* 2003 and *PLOS One* 2010). Several previous papers have shown that X-ray irradiation is an excellent model to induced hepatocyte senescence (Kang et al. *Science* 2015; Freunde et al. *Mol Biol Cell* 2012; Serra et al. *Int J Radiat Biol* 2014). Therefore, we used it on our isolated adult hepatocytes from mice. In fact, we observed induction of senescence markers including persistent DNA damage foci and Sen- β -GAL in the course of 1 week after irradiation.

7. The authors compare DR with ad libitum fed animals. I wonder why they did not use a high fat diet. Rats fed a HF diet develop marked steatosis and I assume this is true for mice as well. Alternatively, or in parallel, they could have used ob/ob mice, which develop spontaneous hepatic steatosis. The same is true for acyl-coenzyme A oxidase KO mice. This is the rate-limiting enzyme of peroxisomal β -oxidation of LCFA, so this could have been a way to directly test the involvement of senescent cells. In such KO mice you would not expect a relationship with senescent cells unless this condition would in fact drive cellular senescence.

We thank the reviewer for the excellent suggestions. We have now done the following experiments: we exposed INK-ATTAC mice to high-fat diet, followed by specific elimination of senescent cells using the AP drug. We found that AP rescued the increase in senescent cells induced by high-fat diet and reduced hepatocyte fat deposits significantly (Fig. 3e-m). Furthermore, we treated db/db mice with senolytic drugs (D+Q). Similarly, we found a reduction in senescent markers as well as fat deposits in these animals (Supp. Fig. 3k-m). We agree with the reviewer that these experiments strengthen the evidence for a relationship between cell senescence and fat accumulation.

Reviewer #2 (Expert in liver biology; Remarks to the Author):

Below, I summarize my responses to the key points that the Editors want reviewers to address:

A. Summary of key results: The data clearly show that hepatocytes in steatotic livers are senescent and that deleting senescent cells reduces steatosis.

B. Originality and interest: The concept that hepatocytes in fatty livers are senescent is not new (over a decade ago, references demonstrated increased p21 in steatotic hepatocytes of genetically obese mice and suggested that hepatocyte senescence contributed to the impaired regenerative capacity of fatty livers). The current work does present data which support the concept that senescence could also cause hepatic steatosis (rather than simply being a consequence of steatosis). However, that evidence is indirect (i.e., acquired in cultured fibroblasts rather than hepatocytes), making it impossible to determine if the mechanisms are cell-autonomous (i.e., autocrine effects in hepatocytes) or non cell-autonomous (i.e., paracrine interactions between hepatocytes and other cell types).

We appreciate the insightful comments from the reviewer. We do not claim in the manuscript that we are the first to propose the concept of hepatocyte senescence or its association to fatty liver, however, only now we have the genetic tools to explore the causal relationships between senescence and fatty liver disease. Using the INK-ATTAC mouse model, we demonstrate that specific clearance of senescent cells reduces fat deposits in hepatocytes from ageing and obese mice. We agree with the reviewer and we have discussed accordingly that we cannot determine, if the mechanisms are cell autonomous from the clearance experiments. However, we have now induced senescence in cultured hepatocytes- where we show senescence-induction occurring concomitantly with increased fat deposition and reduced fatty acid oxidation. Moreover, we have induced senescence specifically in hepatocytes *in vivo* (by impairment of DNA repair, Fig. 4) and show that this results in increased steatosis. Finally, we show a potential mechanism (senescence-associated mitochondrial dysfunction, specifically decreased capacity for FA oxidation) now operating in multiple senescent cell types including hepatocytes.

C. Data and methodology: Multiple complementary approaches were used to demonstrate that hepatocytes in fatty livers are senescent and so this conclusion is assured. However, the studies that attempt to assess causality have significant limitations: a) the mechanistic experiments that assessed effects on fat metabolism were done in cultured fibroblasts rather than hepatocytes (mechanistic experiments have now been fully repeated in hepatocytes, see Fig. 5) and b) the *in vivo* approaches used to eliminate senescent cells are not selective for hepatocytes, making it impossible to determine if hepatic steatosis improved simply because senescent hepatocytes were deleted or if the changes in liver fat reflected changes in the senescence of some other

type of liver cell (e.g., liver fibroblasts) or cells outside of the liver (e.g., adipocytes). A hepatocyte-selective approach has been performed *in vivo* (Fig. 4).

D. Conclusions: The evidence that globally reducing senescence improves steatosis is unequivocal. However, the mechanisms involved remain obscure (see above). Also, the significance of reducing hepatic fat is uncertain in human NAFLD as natural history studies have consistently demonstrated that the severity of hepatic steatosis does not predict meaningful endpoints in human NAFLD (i.e., overall mortality, liver-related mortality, liver-related morbidity, risk of NASH, risk of liver fibrosis). These results challenge the conclusion that anti-senescent therapies would be helpful for preventing NAFLD-related morbidity/mortality in humans.

We believe that with the additional experiments we have now sufficiently clarified that hepatocyte senescence can cause steatosis as well as the mechanism responsible for it (see above).

We completely agree with the reviewer that hepatic steatosis on its own is insufficient as predictor of NAFLD-related mortality and morbidity. Accordingly, we have moderated our conclusions regarding the causal relationships between steatosis and NAFLD progression. Visible steatosis may be considered as a surrogate for the invisible fatty acid flux that potentially drives transition from steatosis to NASH and alternatively as an inert buffering compartment to park some of the fatty acid flux. Thus, more steatosis may be both an indicator of a greater threat to health and evidence of a better safety response. We observed an increase in steatosis with senescence, which may reflect either a greater flux (bad), or greater buffering (good). However, our intervention to ameliorate senescent cells does not promote increased damage (e.g. TAF, p21 etc.) so it's reasonable to assume that greater flux is the predominant driver of steatosis in this model and that interventions to clear senescent cells may bring beneficial effects by reducing it.

Interestingly, published as well as yet unpublished evidence shows that ablation of senescent cells improves whole body metabolism, rescues excessive generation of Reactive Oxygen Species and reduces chronic systemic inflammation, factors that are relevant predictors of NAFLD severity.

E. Suggested improvements:

1) Test effect of senescence on lipid metabolism in hepatocytes directly - this is necessary to conclude that senescence acts in a cell-autonomous fashion to cause hepatocyte lipid accumulation.

We have done it (Fig. 5).

2) Determine if the secretome of fatty hepatocytes stimulates other cells/organs to produce factors that induce senescence in fatty hepatocytes. Such experiments are critical if the studies suggested in #1 yield negative results.

Our results related to #1 above demonstrated clearly cell-autonomous induction of hepatocyte lipid accumulation by hepatocyte senescence both in vivo and in vitro. We therefore would like to perform the additional experiments suggested here as part of a later study focussing on systemic effects of senescent cell ablation.

3) Quantify cell type-specific changes in various types of liver cells across the spectrum of NAFLD to determine if/how changes in senescence correlate with progression of liver injury and fibrosis. This work is necessary to judge the ultimate worth of inhibiting cellular senescence in NAFLD patients.

We find neither in ageing mice nor in liver biopsies from NAFLD patients of variable severity, strong evidence for senescence in cells other than hepatocytes (see response to editor's comment 3 above). These observations might be limited by the low frequencies of some of the non-hepatocyte cell types, however, we did consult an experienced liver pathologist for the assessment of senescence (by p21 staining) in the patient biopsies. Furthermore, we have not seen any fat in other cell types than hepatocytes in mice and human (assessed by our pathologist).

Reviewer #3 (Expert in senescence; Remarks to the Author)

Ogrodnik et al describe how cellular senescence acts as a driver of age-dependent hepatic steatosis. The authors analyse age-dependent hepatic steatosis and their first observation is that dietary restriction can prevent and reverse the accumulation of fat in the liver. They observe a correlation between accumulation of cells with what they describe as markers of senescence and steatosis. Using both genetic models to eliminate senescent cells and drug treatment with specificity to kill senescent cells, the authors show that ablating senescent cells results in reduced fat accumulation. By analysing RNASeq data and performing experiments with mouse fibroblasts they suggest that mitochondria in senescent cells lose the ability to metabolise fatty acids and as a consequence accumulate lipid droplets. The study suggests that senescence is a cause driving hepatic steatosis. While this is an interesting suggestion that broadens the impact of senescence in age-associated disease, there are a number of concerns.

Major points

1. The authors define senescent cells using as main reference the numbers of TAFs and SADs per hepatocyte, in addition they observe a correlation with 'karyomegalic hepatocytes' and p-p38 and 4HNE. While it is plausible that the authors are identifying senescent cells the markers that they use are not what any study will identify as standard markers for senescence. Ideally the authors should present data on SA-b-gal staining and p16 positive cells. While p16 staining can be technically complicated, some experiments have been performed in the INK-ATTAC mice that would allow for measurement of GFP or the INK-ATTAC transgene. There are two concerns here: that standard markers of senescence are not being used, and that the identification of SADs

or TAFS in tissue sections is technical difficult, difficult to reproduce and prone to artefacts as it will require to analyse co-localisation of FISH and IF signals (or measure of distances) using confocal of the tissue slices. Also, how many nuclei have been quantified per slide? How many slides per mice?

We thank the reviewer for these important points. We have now performed both p16 staining (by RNA ISH, using both native p16 and the eGFP reporter in INK-ATTAC mice) and Sen- β -Gal staining with identical results to our previously used techniques, showing in each case the same direction of changes. In our hands, we consistently observe lower frequencies of Sen- β -Gal-positive cells in comparison to all other markers of senescence. This may be related to the known low sensitivity of this histochemical stain in mouse tissues.

Our labs have long-standing expertise in analysing TAF in tissue sections. We perform routinely blinded analyses of Z stacks with 100 nuclei per condition (and did so for all analyses in the present ms). Data here were verified by blinded analysis from 2 independent researchers and we find it highly reproducible amongst them. In fact, we find TAF analysis the most quantitatively reproducible senescence marker (Jurk et al, 2014; Birch et al, 2015; Correia-Melo et al. 2016). 2 sections per mouse were quantified.

2. The authors refer to senescence hepatocytes. Are other cell types (such as stellate cells and others) contributing to the senescence population in aged livers?

We focussed on hepatocytes because they are by far the most ubiquitous cell type in the liver and they are the only cell type to accumulate the fat (this was assessed independently by a liver pathologist) even if only senescence was induced in hepatocytes (Fig. 4 and 5). As shown above (see response to editor), we do not see much evidence for senescence in non-hepatocytes, and have therefore not attempted to further sub-divide these cells.

3. One of the key points of the manuscript is that the elimination of senescent cells decreases fat accumulation in liver. One of the things that is surprising on the experiment presented in Figure 3 are the numbers used and the results obtained given the known intrinsic variability of weight and fat accumulation in B1/6 mice. In the data presented in Figure 3 the size of the 3 groups analysed (control, D+Q and AP) varies between panel. For example for the control group there are 7.7.4.5 or 6 mice analysed. Why this variation? More intriguingly panel d see statistically significant differences using n=4 in the control group. Given the variation and the fact that the highest scoring mice in the control group is driving the significance I am very wary of over interpreting results that seem to be significantly underpowered

We apologise for this. We have now included additional mouse liver sections in all histological assessments, i.e. 6 mice/group for controls and AP, D+Q mice (Fig. 3).

4. Previous work from the van Deursen lab using the INK-ATTAC mice suggests that repeated treatment with AP to deplete senescent cells is successful in many organs but not in liver. The data presented here suggest that 3 single doses of AP are enough to

have an effect on eliminating senescent cells in liver. These discrepancies need to be explained.

This is an excellent point and we are aware of the discrepancy. One issue could be that the van Deursen lab used whole liver RT-PCR to analyse expression of p16. In agreement their data, we did not observe significant differences for p16 by RT-PCR (not shown), however, we consistently detected very low p16 mRNA levels in liver, which suggests that the method may not be sensitive enough to discriminate between treatments. For that reason, we have conducted RNA-ISH. Using RNA-ISH against p16 and eGFP we found that AP treatment was sufficient to suppress the increase in p16 positive cells induced by high fat diet (and this correlated very well with other markers such as TAF and karyomegaly) (Supp. Fig. 3g, h). Furthermore, we would like to apologize to the reviewer for a mistake in material and methods, which affected the interpretation of the results. 24m old mice were only treated with D+Q once per month for 3 months, while INK-ATTAC mice were treated with AP every 3 days for 3 months. We have now made sure that this was corrected in the manuscript.

5. D+Q have been described as a senolytic drug by the Kirkland lab that participates in the study. Senolytic drugs are of great interest. While ABT-263 and ABT-737 have been used by multiple groups for eliminating senescent cells, this is not the case for D+Q. It would be interesting that the authors expand on the results to prove the efficacy of D+Q to eliminate senescent hepatocytes in the present study.

We show that D+Q is as effective as INK-ATTAC/AP to reduce TAF-positive cells in old mice (Fig. 3b). D+Q also reduced TAF-positive hepatocytes to background levels in db/db mice. We validated this marker against two different measures of p16 (Supp. Fig. 3 g, h). D+Q also tended to reduce further senescence markers (karyomegalic cells and cells harbouring DNA damage foci, Fig. 3c and Supp. Fig. 3c).

6. The authors suggest that mitochondrial dysfunction on senescent cells is responsible of reduced fatty acid oxidation capacity and explain fat accumulation on senescent cells. The authors show just correlative studies in fibroblasts and do not perform decisive manipulations to show that this and not other mechanism explain the accumulation of fat in senescent cells.

Our additional experiments (Fig. 4 and 5) now show that senescence-associated mitochondrial dysfunction in hepatocytes is sufficient to induce fat accumulation in senescent liver cells. This does not exclude the possibility, as the reviewer rightly pointed out, that other, especially cell-non-autonomous or systemic mechanisms could also contribute to steatosis, and we acknowledge this possibility in our discussion. However, we think we have now stringently proven the existence of a cell-autonomous mechanism, and believe that proving or disproving alternatives would be the topic for a separate paper(s).

7. In the in vitro studies the authors use fibroblast. Do senescent fibroblasts (or stellate cells) accumulate in old mice showing fat deposits?

We have repeated all *in vitro* experiments in isolated hepatocytes and our pathologist has not seen any fat in other cell types than hepatocytes.

8. Senescent cells accumulate lysosome and possible other autofluorescent vesicles. Although Nile red should be a specific dye to identify lipids, additional controls or parallel methods should be used to discard artefacts.

We thank the reviewer for this excellent point. We have now conducted BODIPY® 493/503 staining in cultured hepatocytes which led to similar results as Nile Red (Supp. Fig. 5 a-d). Liver steatosis was graded by a pathologist (Supp. Fig.1b) and stained with Oil Red and Nile Red which showed similar results.

9. The RNASeq data suggest that genes grouped under the ontology term of 'lipid modificaton' are upregulate during ageing. It would be interesting to link some of the specific genes identified there to the mechanism.

We thank the reviewer for the suggestion. We agree it is an interesting topic, albeit a complex one and we are planning to follow this up in a separate manuscript.

Reviewers' Comments:

Reviewer #1:

Remarks to the Author:

I commend the authors for doing a very good job in revising their manuscript. I am completely satisfied and they addressed all my concerns satisfactorily. I think this is an important contribution to the literature.

Reviewer #2:

Remarks to the Author:

The authors have done an excellent job responding to the critique and revising the manuscript in accordance with new data that were generated to address reviewer questions/concerns.

Briefly,

1 - Several new results have been provided to support the initial suspicion that cellular senescence is largely confined to hepatocytes in this model.

2 - Additional new data demonstrate that hepatocyte senescence induction (by causing hepatocyte DNA damage) tightly parallels hepatic lipid accumulation, strengthening the contention that hepatocyte senescence promotes hepatocyte steatosis. Further, a potential mechanism is suggested by the expression profiling data which document senescence associated changes in lipid homeostatic pathways.

3 - The authors have also appropriately softened their statements about the link between steatosis, senescence and NAFLD progression to acknowledge that their work has not examined the relationship between these outcomes and liver fibrosis.

Incorporation of these new findings have strengthened the manuscript.

Reviewer #3:

Remarks to the Author:

Ogrodnik et al have tried to address most of the comments raised by the reviewers and summarised by the Editors.

Although the present revision improves or try to improve many of the criticisms there are still questions that remain:

1. The authors have made a better effort to define 'senescence' in hepatocytes by expanding their use of markers and using markers that would be recognised by the general community as senescence markers. While p16 staining in mouse tissues is problematic, the authors make a good effort and substitute it with p16 ISH. However, it is difficult to understand how to put together their comment on low levels of p16 detected by qRT-PCR in the liver, with more than 20% of p16 positive hepatocytes when they measure it by p16ISH.

2. The authors talk in their rebuttal about the discrepancy between the inability of the van Deursen lab to validate the INK-ATTAC transgene as functional in liver and the results they describe here. However their argument of low levels of p16 as detected by qRT-PCR is difficult to square with the discrepancies in between the labs; the effects they observe in liver (and the ISH results).

3. The authors respond to the criticism of the studies being underpowered and showing differences in the mice included per group or per experiment adding mice up to 6 per group. It is still surprising that given the inherent variability of phenotypes such as steatosis that is enough to see significant differences.

4. The authors carry out experiments in vitro using hepatocytes that are analysed after senescence is induced with irradiation. Hepatocytes are kept in culture for a week. Are the control hepatocytes ok after a week? From my knowledge and experience mouse hepatocytes cannot be maintained in culture without significance caveats (to their viability, cell fate and senescence) for so long. The authors should give more details on how did they culture them and assess their 'integrity'.

1. The authors have made a better effort to define 'senescence' in hepatocytes by expanding their use of markers and using markers that would be recognised by the general community as senescence markers. While p16 staining in mouse tissues is problematic, the authors make a good effort and substitute it with p16 ISH. However, it is difficult to understand how to put together their comment on low levels of p16 detected by qRT-PCR in the liver, with more than 20% of p16 positive hepatocytes when they measure it by p16ISH.
2. The authors talk in their rebuttal about the discrepancy between the inability of the van Deursen lab to validate the INK-ATTAC transgene as functional in liver and the results they describe here. However their argument of low levels of p16 as detected by qRT-PCR is difficult to square with the discrepancies in between the labs; the effects they observe in liver (and the ISH results).

We have now shown several markers of senescence, including p21, TAF, karyomegaly and SA- β -Gal and we show correlations between different markers (Supp. Fig. 3g, h). Furthermore, we have verified that p16 and eGFP staining show similar values and that staining on adjacent sections is located in the same cells/areas (Supp. Fig. 3i). We and others groups have tried to measure p16 in liver with qRT-PCR and we have not been successful (even in older animals 28months +). Importantly, our assays are at single cell level, while by PCR not only both senescent and non-senescent hepatocytes but also all other cell types (endothelia cells, Kupffer cells, Stellate cells, epithelia cells, fibroblasts and immune cells) contribute to the end result. A simple calculation shows that qPCR is not able to detect these changes:

Our markers indicate changes in frequencies of senescent hepatocytes from about 3 to 8% between DR and AL at 15mo of age and between 5% and 15-20% in old or obese mice. Let's take the largest difference, between 5 and 20%. Numerically, hepatocytes are about 60% of all liver cells. Assuming that the level of p16MRNA in a senescent cell is n times that of a young cell, we get

$$p16mRNA \text{ (old tissue)} = p16mRNA \text{ (young cell)} \times [(0.2n + 0.8) \times 0.6 + 0.4]$$

$$p16mRNA \text{ (young tissue)} = p16mRNA \text{ (young cell)} \times [(0.05n + 0.95) \times 0.6 + 0.4]$$

Assuming a reasonable $n=3$, the ratio between the tissues is $[(0.6 + 0.8) \times 0.6 + 0.4] / [(0.15 + 0.95) \times 0.6 + 0.4] = 1.24/1.06 = 1.17$. This is essentially undetectable by qPCR. Even assuming $n=10$, the ratio would only be 1.64, which would still be very hard to detect.

3. The authors respond to the criticism of the studies being underpowered and showing differences in the mice included per group or per experiment adding mice up to 6 per group. It is still surprising that given the inherent variability of phenotypes such as steatosis that is enough to see significant differences.

We refute the implication that we might somehow selected our mice. In the first submission, our selection was completely random and we did see significant differences between groups already then. In response to the reviewer, we included all available mice in the assessments and confirmed significance.

4. The authors carry out experiments in vitro using hepatocytes that are analysed after senescence is induced with irradiation. Hepatocytes are kept in culture for a week. Are the control hepatocytes ok after a week? From my knowledge and experience mouse hepatocytes cannot be maintained in culture without significance caveats (to their viability, cell face and senescence) for so long. The authors should give more

details on how did they culture them and assess their 'integrity'.

We are in agreement with the reviewer that there are technical challenges in culturing primary hepatocytes, but we performed these experiments as requested by the reviewers. We have kept hepatocytes at a more physiological oxygen tension (3%O₂), which allowed us to maintain cells for longer and reduced culture induced cell-death. Following 10Gy X-ray irradiation we observed that hepatocytes acquire a morphology characteristic of senescence and SA-β-Gal activity after 6 days. Monitoring cell numbers revealed that a small percentage of hepatocytes experienced cell-death after irradiation, however, in our culture conditions most of the cells survived and acquired a senescent-like phenotype.

In the non-irradiated controls, we saw changes in viability and cell morphology with time. This was mostly due to overgrowth by cell-types other than hepatocytes (which are present in very low frequencies), such as fibroblasts and endothelial cells. This did not happen after irradiation. In addition, we cannot exclude other events happening to some of the hepatocytes, including culture stress leading to either senescence, apoptosis or transformation. For that reason, control cells were analysed 1 to 2 days following isolation, at the same time as irradiation took place for the irradiated cells.